# FACE-HUMAN-BENCH: A COMPREHENSIVE BENCHMARK OF FACE AND HUMAN UNDERSTANDING FOR MULTI-MODAL ASSISTANTS

## ABSTRACT

Faces and humans are crucial elements in social interaction and are widely included in everyday photos and videos. Therefore, a deep understanding of faces and humans will enable multi-modal assistants to achieve improved response quality and broadened application scope. Currently, the multi-modal assistant community lacks a comprehensive and scientific evaluation of face and human understanding abilities. In this paper, we first propose a hierarchical ability taxonomy that includes three levels of abilities. Then, based on this taxonomy, we collect images and annotations from publicly available datasets in the face and human community and build a semi-automatic data pipeline to produce problems for the new benchmark. Finally, the obtained Face-Human-Bench comprises a development set with 900 problems and a test set with 1800 problems, supporting both English and Chinese. We conduct evaluations over 25 mainstream multi-modal large language models (MLLMs) with our Face-Human-Bench, focusing on the correlation between abilities, the impact of the relative position of targets on performance, and the impact of Chain of Thought (CoT) prompting on performance. Moreover, inspired by multi-modal agents, we also explore which abilities of MLLMs need to be supplemented by specialist models. The data and evaluation code of the Face-Human-Bench will be made publicly available.

## 1 INTRODUCTION

Faces and humans are always the most crucial elements of photos and videos in our everyday lives. Consequently, they are also critical focuses in multi-modal AI applications. In the past two years, ChatGPT (OpenAI, 2023a) and GPT-4 (OpenAI, 2023b) have achieved great success with impressive instruction-following and multi-modal understanding capabilities respectively. Numerous excellent works (Liu et al., 2023b; Zhu et al., 2024; Dai et al., 2023; Bai et al., 2023) from the open-source community have followed, collectively presenting the immense potential of multi-modal assistants. Since faces and humans are central to social interaction, a deep understanding of this information can make multi-modal assistants achieve improved response quality and broadened application scope. For instance, in movie understanding (Yue et al., 2023; Han et al., 2023; Wang et al., 2024), identifying characters is a prerequisite for multi-modal assistants to describe the plot accurately. In multi-modal human-computer interaction (Fu et al., 2024), perceiving expressions and body language can help multi-modal assistants accurately understand the context, generating more personalized and humanized responses. In media forensics (Liu et al., 2024b;c; Jia et al., 2024), determining whether deepfake artifacts exist on a face is crucial for multi-modal assistants to detect misinformation.

Comprehensive and scientific evaluation is the foundation for researching applications of multi-modal assistants related to "faces and humans." Existing benchmarks Fu et al. (2023); Li et al. (2023a); Liu et al. (2023c) for large multi-modal models typically involve limited abilities of face and human understanding, such as celebrity recognition, action recognition, identity reasoning, and social relation, leaving many important abilities unexplored. On the other hand, since face and human understanding is one of the earliest research topics in artificial intelligence, there are numerous datasets available for evaluating the performance of specialist models. The images and annotations from these datasets can serve as original material to evaluate multi-modal assistants.

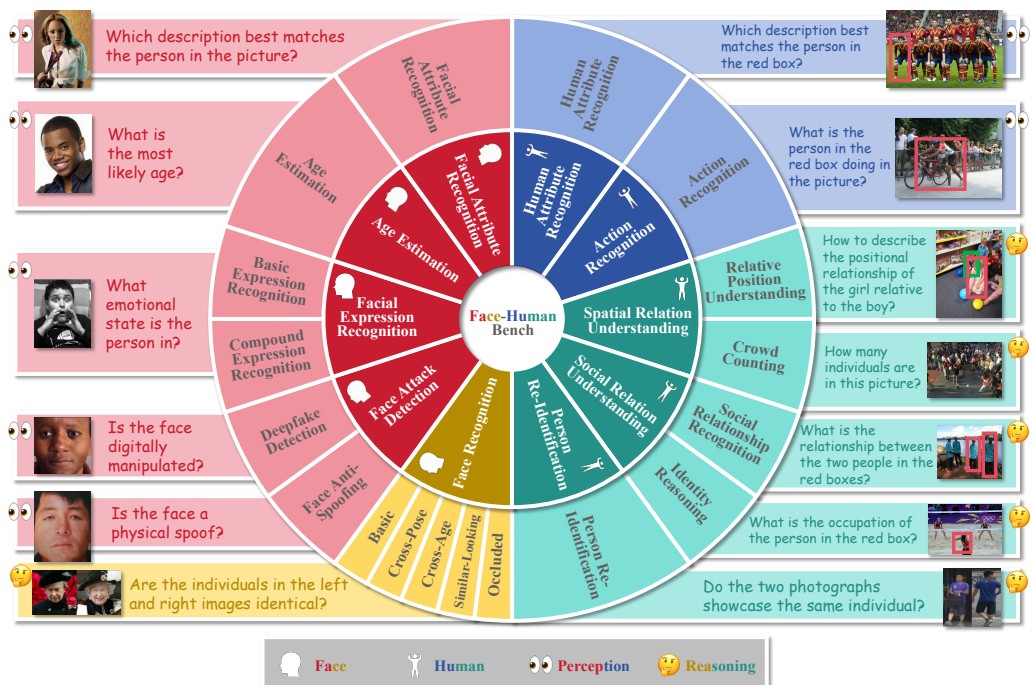

Figure 1: The hierarchical ability taxonomy for evaluating face and human understanding abilities. We construct the Face-Human-Bench based on this taxonomy. The proportion of the sectors represents the weight of the corresponding abilities in the overall score on the Face-Human-Bench.

As the starting point of our evaluation, we propose a hierarchical ability taxonomy, as shown in Figure 1. This taxonomy consists of three levels. Level-1 (L1) has two perspectives to study: from the target perspective, L1 includes face understanding and human understanding; from the cognitive process perspective, L1 includes perception and reasoning. Subsequently, we incorporate finer-grained abilities into the taxonomy and categorize them into 10 Level-2 (L2) and 18 Level-3 (L3) ability dimensions. Then, based on this taxonomy, we collect datasets from the face and human community and use a semi-automatic data pipeline to transform original images and annotations into multi-modal QAs. The final obtained benchmark called Face-Human-Bench, including a development set with 900 problems and a test set with 1800 problems, supporting evaluations in both English and Chinese. For ease of evaluation, we adopt multiple-choice as the problem format following MMBench (Liu et al., 2023c) and SEED-Bench (Li et al., 2023a).

In the literature, multi-modal assistants can be broadly categorized into two types: (1) Multi-modal large language models (MLLMs), which achieve end-to-end output by aligning visual information to the language domain with visual instruction-tuning (Liu et al., 2023b). (2) Multi-modal agents (Wu et al., 2023; Yang et al., 2023), where LLMs decide when to call specialist models to solve particular problems and then integrate the outputs of these specialist models. Compared to multi-modal agents, MLLMs generally have better multi-modal perception and reasoning abilities with more effective relationship modeling across modalities. In this study, the first research question (**RQ1**) is: *"How do existing MLLMs perform in face and human understanding?"* In this question, we focus on (a) the performance of 25 mainstream MLLMs, (b) the correlation between abilities at different levels, (c) the impact of the relative position of targets on performance, and (d) the impact of Chain of Thought (CoT) prompting on performance. Meanwhile, for face and human understanding tasks in which specialist models significantly outperform MLLMs, we can draw inspiration from multi-modal agents by utilizing the output of these specialist models to enhance the responses of multi-modal assistants. Thus, the second research question emerges (**RQ2**): *In the field of face and human understanding, which tasks' specialist models can achieve significantly better performance than current MLLMs?*

In response to RQ1, our main findings are as follows: (a) The Face-Human-Bench effectively distinguishes the abilities of MLLMs in faces and human understanding. Under the zero-shot setting,

the best-performing closed-source model, GPT-4o (OpenAI, 2024), does not perform as well as the best open-source model, InternVL-Chat-v1.2-Plus (Chen et al., 2023). (b) The correlation coefficients can reveal correlations between abilities at different levels. At L2 and L3, there are some ability groups in which the ability dimensions exhibit significant positive correlations between each pair. (c) Many models show substantial performance differences on the same task with different relative positions of targets. We design a new metric called the relative position sensitivity score (RPSS) to measure this phenomenon. On this metric, InternLM-XComposer2-VL-7B (Dong et al., 2024) performs the best, indicating that its performance is almost unaffected by the relative position of targets. (d) Introducing hints and CoT instructions into the prompts significantly improves the performance of the closed-source model GPT-4o, but has no effect on the open-source model, InternVL-Chat-v1.2-Plus. In response to RQ2, we find that in tasks of deepfake detection, crowd counting, and face recognition (under challenging scenarios), the performance of MLLMs is significantly inferior to that of corresponding specialist models. Therefore, we recommend incorporating specialist models in applications requiring these abilities to help improve the response quality of multi-modal assistants.

Our contributions can be summarized as follows:

- We propose the Face-Human-Bench, the first benchmark dedicated to evaluating multimodal assistants' face and human understanding abilities. The Face-Human-Bench is based on a three-level ability taxonomy and supports both English and Chinese.
- Utilizing the Face-Human-Bench, we conduct a comprehensive evaluation of mainstream MLLMs, revealing the correlation between abilities, and exploring the impact of the relative position of targets and CoT prompting on the performance of MLLMs.
- We explore which specialist models significantly outperform MLLMs in certain face and human understanding tasks. Based on this, we provide suggestions for enhancing the response quality of multi-modal assistants.

## 2    FACE-HUMAN-BENCH

### 2.1    HIERARCHICAL ABILITY TAXONOMY

As shown in Figure 1, the proposed ability taxonomy includes three levels. Level-1 (L1) has two research perspectives. From the target perspective, L1 includes face understanding and human understanding. From the cognitive process perspective, L1 includes perception and reasoning. In our evaluation, perception involves direct comprehension of only one target, while reasoning requires synthesizing information from multiple targets and environments to conclude. There are ten abilities in total at Level-2 (L2). Five are focused on faces: facial attribute recognition, age estimation, facial expression recognition, face attack detection, and face recognition, and five are focused on humans: human attribute recognition, action recognition, spatial relation understanding, social relation understanding, and person re-identification. It should be noted that at L2, there are 6 abilities under perception and 4 abilities under reasoning. Level-3 (L3) further refines the ability dimensions at L2. Facial expression recognition can be categorized into basic and compound types. Face attack detection includes deepfake detection and face anti-spoofing. Face recognition involves five scenarios: basic, cross-pose, cross-age, similar-looking, and occluded. Spatial relation understanding concerns relative position and count. Social relation understanding includes social relationship recognition and identity reasoning. Please refer to Appendix A.1 for detailed definitions and examples of these abilities.

### 2.2    SEMI-AUTOMATIC DATA PIPELINE

Based on the hierarchical ability taxonomy defined in Section 2.1, we collect 16 public datasets from the face and human community, covering each L3 ability. Then, we employ a semi-automatic data pipeline to produce problems for the Face-Human-Bench.

An original sample $S_i$ from public datasets can be represented as a binary tuple $(I_i, L_i)$, where $I_i$ denotes an original image set and $L_i$ denotes an original label set. Note that we use "image set" and "label set" to describe the composition of one sample because, in some datasets, a single sample may consist of multiple images or labels. For instance, in face recognition, a sample includes a pair of

face images to verify identity, and in facial attribute recognition, a sample may involve 40 attribute labels.

For ease of evaluation, we adopt multiple-choice as the problem format in our Face-Human-Bench. Each problem $P_i$ corresponds to a quadruple $(V_i, Q_i, O_i, A_i)$. Here, $V_i$ refers to the images obtained via the image processing pipeline $p_{image} : \mathbb{I} \rightarrow \mathbb{V}$. $p_{image}$ performs an operation such as cropping, concatenating, adding boxes, or leaving the original images unchanged, depending on the ability to test. $Q_i$ denotes the question. Each L3 ability includes a set of pre-written questions that share the same semantics but exhibit diversity. When producing samples, a question $Q_i$ is randomly selected from this question set. $O_i$ is the set of $n$ options $(o_1, o_2, ..., o_n)$, where $2 \leq n \leq 4$. These options are obtained through the text processing pipeline $p_{text} : \mathbb{L} \rightarrow \mathbb{O}$. $p_{text}$ converts the original labels into one correct option and $n - 1$ incorrect options. For some tasks, ChatGPT (OpenAI, 2023a) is used within $p_{text}$ to assist in generating incorrect options or adjusting options at the sentence level (fixing grammar or re-wording sentences for fluency. $A_i$ is the correct answer to the problem. The produced $P_i$ will be checked by data reviewers to ensure that the options are unambiguous and there is one and only one correct answer. The problems that do not meet the requirements will be removed.

In summary, our semi-automatic data pipeline leverages image and text processing pipelines, $p_{image}$ and $p_{text}$, to transform original samples into multiple-choice format problems. These problems are then manually checked to ensure quality. We obtain a benchmark with a development set of 900 problems for the MLLM community to evaluate during training iterations and a test set of 1800 problems for the formal evaluation in our paper. Additionally, the English problems are translated into Chinese to create a Chinese version of the benchmark. For more details on data sources, statistics, and the semi-automatic data pipeline, please refer to Appendices A.2 and A.3.

## 3 EXPERIMENT

### 3.1 EXPERIMENTAL SETUP

**Evaluation Protocols.** We use the weighted accuracy of multiple-choice problems as the evaluation score. As shown in Figure 1, the proportion of the sectors represents the weight of the corresponding abilities in the overall score on the Face-Human-Bench. Note that we set equal weights for each L2 ability. [1] To prevent models from favoring certain option letters over others, we shuffle the options to ensure the correct answers are evenly distributed across all option letters. During the testing, we add some constraint instructions to ensure MLLMs output only option letters as much as possible. [2] After obtaining the MLLM's response, we use regular expressions to extract the option letters. If this fails, we follow the implementation of MMBench (Liu et al., 2023c) using ChatGPT (OpenAI, 2023a) to extract the choices. [3]

**Models.** We evaluate 25 MLLMs in different sizes from 13 model families. For open-source models, we select LLaVA-13B (Liu et al., 2023b), LLaVA-1.5-7B/13B (Liu et al., 2023a), LLaVA-Next-7B/13B/34B (Liu et al., 2024a), MiniGPT-4-7B/13B (Zhu et al., 2024), InstructBLIP-7B/13B (Dai et al., 2023), Qwen-VL-Chat (Bai et al., 2023), InternLM-XComposer2-VL-7B (Dong et al., 2024), Yi-VL-6B (Young et al., 2024), InternVL-Chat-v1.2-Plus (Chen et al., 2023), InternVL-Chat-v1.5 (Chen et al., 2023), DeepSeek-VL-1.3B/7B-Chat (Lu et al., 2024), CogVLM2-19B-Chat (Hong et al., 2024), GLM-4V-9B (Hong et al., 2024), LLaVA-OneVison-0.5B/7B (Li et al., 2024). For closed-source models, we use Gemini-1.5-Pro (Reid et al., 2024), Claude-3.5-Sonnet (Anthropic, 2024a), GPT-4V (OpenAI, 2023b), and GPT-4o OpenAI (2024). For more details on these models, please refer to Appendix B.1.

### 3.2 MAIN RESULTS

Table 1 shows the performance of all evaluated MLLMs at different levels of abilities on the Human-Face-Bench (English) [4] under the zero-shot setting. Overall scores range from 27.9% to 76.4%, demonstrating the effectiveness of the Face-Human-Bench in distinguishing the abilities of MLLMs

---

[1] For detailed weights of each subset in Face-Human-Bench, please refer to Appendix A.2.

[2] For the prompt template under zero-shot setting, please refer to Appendix B.2.1.

[3] For the prompt for choice extraction, please refer to Appendix B.2.2.

[4] For the results of the Chinese version, please refer to Appendix C.2.

Table 1: Zero-shot scores of MLLMs on the hierarchical Face-Human-Bench (EN). The highest scores for open-source and closed-source MLLMs are marked in blue and green respectively.

| Model | | | Face Understanding | | | | | | | | | | |
|---|---|---|---|---|---|---|---|---|---|---|---|---|---|
| | Attr. | Age | Expression | | | Attack Detection | | | Face Recognition | | | | | |
| | | | Basic | Comp. | Mean | DFD | FAS | mean | Basic | C.P. | C.A. | S.L. | Occ. | Mean |
| Random | 25.0 | 25.0 | 25.0 | 25.0 | 25.0 | 50.0 | 50.0 | 50.0 | 50.0 | 50.0 | 50.0 | 50.0 | 50.0 | 50.0 |
| LLaVA-OneVision-0.5B | 36.0 | 43.0 | 71.0 | 60.0 | 65.5 | 46.0 | 55.0 | 50.5 | 50.0 | 42.0 | 44.0 | 50.0 | 38.0 | 44.8 |
| DeepSeek-VL-1.3B-Chat | 36.5 | 49.0 | 57.0 | 50.0 | 53.5 | 50.0 | 50.0 | 50.0 | 50.0 | 50.0 | 50.0 | 50.0 | 50.0 | 50.0 |
| Yi-VL-6B | 75.5 | 51.7 | 65.0 | 52.0 | 58.5 | 34.0 | 43.0 | 38.5 | 50.0 | 48.0 | 48.0 | 50.0 | 44.0 | 48.0 |
| MiniGPT-4-7B | 24.0 | 17.7 | 26.0 | 24.0 | 25.0 | 31.5 | 40.5 | 36.0 | 38.0 | 56.0 | 44.0 | 48.0 | 34.0 | 44.0 |
| InstructBLIP-7B | 39.5 | 36.7 | 38.0 | 40.0 | 39.0 | 50.5 | 53.0 | 51.8 | 52.0 | 58.0 | 48.0 | 52.0 | 54.0 | 52.8 |
| Qwen-VL-Chat | 55.5 | 49.7 | 65.0 | 50.0 | 57.5 | 51.0 | 54.0 | 52.5 | 66.0 | 52.0 | 54.0 | 58.0 | 54.0 | 56.8 |
| DeepSeek-VL-7B-Chat | 57.5 | 52.3 | 68.0 | 58.0 | 63.0 | 46.0 | 53.0 | 49.5 | 54.0 | 52.0 | 50.0 | 48.0 | 50.0 | 50.8 |
| LLaVA-1.5-7B | 61.0 | 49.3 | 62.0 | 58.0 | 60.0 | 55.5 | 55.0 | 55.3 | 54.0 | 52.0 | 50.0 | 56.0 | 50.0 | 52.4 |
| LLaVA-NeXT-7B | 69.5 | 50.0 | 72.0 | 62.0 | 67.0 | 59.5 | 58.5 | 59.0 | 62.0 | 50.0 | 48.0 | 56.0 | 50.0 | 53.2 |
| InternLM-XComposer2-VL-7B | 92.0 | 53.0 | 76.0 | 68.0 | 72.0 | 41.0 | 54.0 | 47.5 | 54.0 | 54.0 | 50.0 | 56.0 | 36.0 | 50.0 |
| LLaVA-OneVision-7B | 90.5 | 60.3 | 74.0 | 62.0 | 68.0 | 35.0 | 56.0 | 45.5 | 58.0 | 42.0 | 34.0 | 42.0 | 34.0 | 42.0 |
| CogVLM2-19B-Chat | 75.0 | 57.3 | 71.0 | 70.0 | 70.5 | 37.0 | 51.0 | 44.0 | 66.0 | 36.0 | 44.0 | 46.0 | 48.0 | 48.0 |
| GLM-4V-9B | 79.5 | 55.7 | 79.0 | 74.0 | 76.5 | 46.0 | 50.0 | 48.0 | 68.0 | 54.0 | 54.0 | 62.0 | 52.0 | 58.0 |
| MiniGPT-4-13B | 20.5 | 24.3 | 35.0 | 26.0 | 30.5 | 49.5 | 37.5 | 43.5 | 52.0 | 46.0 | 42.0 | 46.0 | 48.0 | 46.8 |
| InstructBLIP-13B | 25.5 | 38.3 | 50.0 | 42.0 | 46.0 | 57.5 | 52.0 | 54.8 | 48.0 | 52.0 | 52.0 | 50.0 | 52.0 | 50.8 |
| LLaVA-13B | 32.0 | 40.7 | 56.0 | 30.0 | 43.0 | 55.0 | 54.0 | 54.5 | 52.0 | 60.0 | 52.0 | 40.0 | 52.0 | 51.2 |
| LLaVA-1.5-13B | 75.5 | 58.7 | 72.0 | 54.0 | 63.0 | 51.0 | 54.0 | 52.5 | 54.0 | 48.0 | 54.0 | 48.0 | 50.0 | 50.8 |
| LLaVA-NeXT-13B | 77.5 | 46.7 | 71.0 | 52.0 | 61.5 | 51.0 | 54.0 | 52.0 | 58.0 | 54.0 | 54.0 | 56.0 | 56.0 | 55.6 |
| InternVL-Chat-v1.5 | 92.0 | 61.7 | 72.0 | 68.0 | 70.0 | 71.5 | 67.0 | 69.2 | 90.0 | 60.0 | 60.0 | 60.0 | 52.0 | 64.4 |
| LLaVA-NeXT-34B | 95.0 | 58.7 | 80.0 | 62.0 | 71.0 | 63.5 | 60.5 | 62.0 | 92.0 | 70.0 | 70.0 | 72.0 | 56.0 | 72.0 |
| InternVL-Chat-v1.2-Plus | 86.0 | 59.7 | 74.0 | 60.0 | 67.0 | 65.5 | 65.0 | 65.3 | 94.0 | 74.0 | 62.0 | 72.0 | 52.0 | 70.8 |
| Gemini-1.5-Pro | 66.0 | 40.0 | 72.0 | 48.0 | 60.0 | 31.0 | 21.0 | 26.0 | 98.0 | 82.0 | 86.0 | 90.0 | 72.0 | 85.6 |
| Claude-3.5-Sonnet | 83.5 | 54.0 | 73.0 | 32.0 | 52.5 | 55.0 | 45.0 | 50.0 | 92.0 | 64.0 | 76.0 | 74.0 | 66.0 | 74.4 |
| GPT-4V | 77.5 | 53.7 | 75.0 | 48.0 | 61.5 | 50.5 | 58.5 | 54.5 | 96.0 | 72.0 | 92.0 | 82.0 | 64.0 | 81.2 |
| GPT-4o | 77.0 | 61.0 | 83.0 | 62.0 | 72.5 | 53.0 | 64.0 | 58.5 | 96.0 | 72.0 | 74.0 | 76.0 | 50.0 | 73.6 |

| Model | | | Human Understanding | | | | | | | | | | | |
|---|---|---|---|---|---|---|---|---|---|---|---|---|---|---|
| | Attr. | Action | Spatial Relation | | | Social Relation | | | Re-ID | Face | Human | Per. | Rea. | Overall |
| | | | RPU | CC | Mean | SRR | IR | Mean | | | | | | |
| Random | 25.0 | 25.0 | 25.0 | 25.0 | 25.0 | 25.0 | 25.0 | 25.0 | 50.0 | 35.0 | 30.0 | 29.2 | 37.5 | 32.5 |
| LLaVA-OneVision-0.5B | 47.0 | 78.0 | 44.0 | 22.7 | 33.3 | 62.0 | 94.0 | 78.0 | 45.0 | 48.0 | 56.3 | 53.3 | 50.3 | 52.1 |
| DeepSeek-VL-1.3B-Chat | 40.5 | 66.0 | 40.0 | 26.0 | 33.0 | 64.0 | 72.0 | 68.0 | 50.0 | 47.8 | 51.5 | 49.3 | 50.3 | 49.7 |
| Yi-VL-6B | 67.0 | 73.0 | 54.0 | 24.0 | 39.0 | 48.0 | 66.0 | 57.0 | 47.0 | 54.4 | 56.6 | 60.7 | 47.8 | 55.5 |
| MiniGPT-4-7B | 15.5 | 27.0 | 18.0 | 16.7 | 17.3 | 24.0 | 34.0 | 29.0 | 44.0 | 29.3 | 26.6 | 24.2 | 33.6 | 27.9 |
| InstructBLIP-7B | 31.0 | 46.0 | 34.0 | 0.7 | 17.3 | 16.0 | 28.0 | 22.0 | 51.0 | 43.9 | 33.5 | 40.7 | 35.8 | 38.7 |
| Qwen-VL-Chat | 49.5 | 83.0 | 54.0 | 34.0 | 44.0 | 64.0 | 70.0 | 67.0 | 50.0 | 54.4 | 58.7 | 57.9 | 54.5 | 56.5 |
| DeepSeek-VL-7B-Chat | 64.0 | 78.0 | 52.0 | 35.3 | 43.7 | 70.0 | 76.0 | 73.0 | 57.0 | 54.6 | 63.1 | 60.7 | 56.1 | 58.9 |
| LLaVA-1.5-7B | 62.0 | 71.0 | 54.0 | 30.0 | 42.0 | 68.0 | 78.0 | 73.0 | 63.0 | 55.6 | 62.2 | 59.8 | 57.6 | 58.9 |
| LLaVA-NeXT-7B | 62.0 | 80.0 | 62.0 | 24.7 | 43.3 | 62.0 | 86.0 | 74.0 | 56.0 | 59.7 | 63.1 | 64.6 | 56.6 | 61.4 |
| InternLM-XComposer2-VL-7B | 87.5 | 87.0 | 58.0 | 41.3 | 49.7 | 64.0 | 86.0 | 75.0 | 59.0 | 62.9 | 71.6 | 73.2 | 58.4 | 67.3 |
| LLaVA-OneVision-7B | 90.5 | 92.0 | 58.0 | 48.0 | 53.0 | 66.0 | 86.0 | 76.0 | 61.0 | 61.3 | 74.5 | 74.5 | 58.0 | 67.9 |
| CogVLM2-19B-Chat | 70.5 | 93.0 | 68.0 | 33.3 | 50.7 | 74.0 | 92.0 | 83.0 | 56.0 | 59.0 | 70.6 | 68.4 | 59.4 | 64.8 |
| GLM-4V-9B | 85.5 | 94.0 | 62.0 | 32.0 | 47.0 | 68.0 | 88.0 | 78.0 | 67.0 | 63.5 | 74.3 | 73.2 | 62.5 | 68.9 |
| MiniGPT-4-13B | 19.5 | 46.0 | 42.0 | 17.3 | 29.7 | 30.0 | 50.0 | 40.0 | 48.0 | 33.1 | 36.6 | 30.7 | 41.1 | 34.9 |
| InstructBLIP-13B | 33.5 | 71.0 | 38.0 | 28.0 | 33.0 | 52.0 | 86.0 | 69.0 | 51.0 | 43.1 | 51.5 | 44.9 | 51.0 | 47.3 |
| LLaVA-13B | 27.0 | 66.0 | 36.0 | 30.7 | 33.3 | 38.0 | 76.0 | 57.0 | 55.0 | 44.3 | 47.7 | 43.9 | 49.1 | 46.0 |
| LLaVA-1.5-13B | 60.5 | 72.0 | 44.0 | 26.0 | 35.0 | 60.0 | 60.0 | 60.0 | 54.0 | 60.1 | 56.3 | 63.7 | 50.0 | 58.2 |
| LLaVA-NeXT-13B | 69.5 | 74.0 | 46.0 | 28.0 | 37.0 | 58.0 | 70.0 | 64.0 | 63.0 | 58.7 | 61.5 | 63.5 | 54.9 | 60.1 |
| InternVL-Chat-v1.5 | 89.5 | 89.0 | 62.0 | 50.7 | 56.3 | 70.0 | 74.0 | 72.0 | 77.0 | 71.5 | 76.8 | 78.6 | 67.4 | 74.1 |
| LLaVA-NeXT-34B | 91.5 | 88.0 | 64.0 | 59.3 | 61.7 | 64.0 | 86.0 | 75.0 | 88.0 | 71.7 | 80.8 | 77.7 | 74.2 | 76.3 |
| InternVL-Chat-v1.2-Plus | 90.0 | 92.0 | 66.0 | 58.7 | 62.3 | 76.0 | 96.0 | 86.0 | 85.0 | 69.7 | 83.1 | 76.7 | 76.0 | 76.4 |
| Gemini-1.5-Pro | 50.0 | 75.0 | 52.0 | 25.3 | 38.7 | 74.0 | 84.0 | 79.0 | 82.0 | 55.6 | 64.9 | 52.8 | 71.3 | 60.3 |
| Claude-3.5-Sonnet | 71.5 | 90.0 | 54.0 | 42.7 | 48.3 | 74.0 | 80.0 | 77.0 | 74.0 | 62.9 | 72.2 | 70.0 | 68.4 | 67.5 |
| GPT-4V | 73.0 | 78.0 | 38.0 | 71.3 | 54.7 | 68.0 | 84.0 | 76.0 | 83.0 | 65.7 | 72.9 | 66.4 | 73.7 | 69.3 |
| GPT-4o | 63.5 | 81.0 | 50.0 | 58.7 | 54.3 | 66.0 | 94.0 | 80.0 | 79.0 | 68.5 | 71.6 | 68.9 | 71.7 | 70.0 |

in face and human understanding. We visualize the overall scores of MLLMs in Figure 2. Our findings can be summarized as follows.

**Overall Performance.** (1) The top three performing open-source models in terms of the overall score are InternvL-Chat-v1.2-Plus, LLaVA-Next-34B, and InternVL-Chat-v1.5. These models' LLMs have the largest number of parameters among all open-source models we evaluate. (2) Generally, open-source models within the same series tend to show improved performance with increasing parameter scale. However, there are exceptions; for instance, the 13B version of LLaVA-1.5

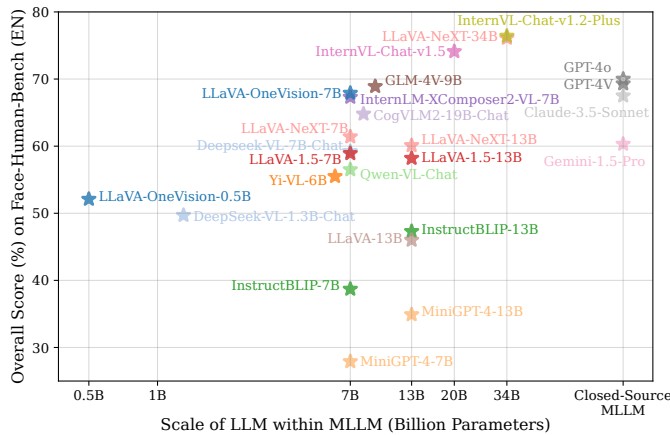

Figure 2: The leaderboard of MLLMs on our proposed Face-Human-Bench (English).

and LLaVA-Next perform slightly worse than their 7B counterparts. (3) Under the zero-shot setting, the best closed-source model, GPT-4o, does not surpass the performance of the top-performing open-source models. We believe this is because GPT-4o does not fully realize its potential under the zero-shot setting. The experiments in Section 3.5 confirm our hypothesis. (4) Newer models show significant improvements compared to earlier models. Among MLLMs with 7B parameters within LLM, the recently released LLaVA-OneVision-7B performs best. Impressively, LLaVA-OneVision-0.5B, with only 0.5B parameters within LLM, outperforms the earlier InstructBLIP-13B.

**L2 and L3 Performance**[5] (1) At L2 and L3, the best performance among open-source models is usually achieved by one of InternvL-Chat-v1.2-Plus, LLaVA-Next-34B, and InternVL-Chat-v1.5. Specifically, GLM-4V-9B achieves the best results in compound expression recognition (L3), facial expression recognition (L2), and action recognition (L2) and CogVLM2-19B-Chat achieves the best result in relative position understanding (L3). (2) At L2 and L3, the best performance among closed-source models is usually achieved by GPT-4o or GPT-4v. Notably, Gemini-1.5-Pro demonstrates outstanding face recognition ability (L2), achieving the best performance among all models with a score of 85.6%.

### 3.3 CORRELATION BETWEEN ABILITIES

In this section, we examine whether improving one ability in a model will enhance another by calculating the Pearson Correlation Coefficient between abilities at different levels, using the evaluation scores from Section 3.2. At L1, the correlation coefficient of face and human understanding is 0.94 and the correlation coefficient of perception and reasoning is 0.79, both indicating significant positive correlations, as shown in Figure 3(a) and Figure 3(b).

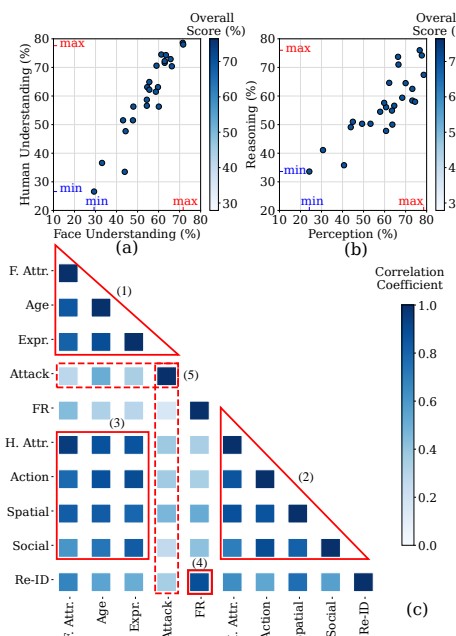

Figure 3: Correlation between abilities.

We further investigate the correlations between L2 abilities, resulting in the correlation coefficient matrix shown in Figure 3(c). For clarity, we have drawn this as a lower triangular matrix. Our findings can be summarized as follows: (1) For the three face understanding abilities—facial attribute recognition, age estimation, and facial expression recognition—there are high positive correlations between each pair. (2) For the four human understanding abilities—human attribute recognition, action recognition, spatial relation understanding, and social relation understanding—there are high positive correlations between each pair. (3) For the three face understanding abilities and four hu-

---

[5]For the visualization of L2 and L3 results, please refer to the Appendix C.1.

man understanding abilities mentioned above, there are high positive correlations between each pair. (4) The two identity recognition tasks—face recognition and person re-identification—show a high positive correlation. (5) The correlation between face attack detection and any other ability is low. In Appendix C.3, we further present the correlations between L3 abilities.

## 3.4 RELATIVE POSITION OF TARGETS

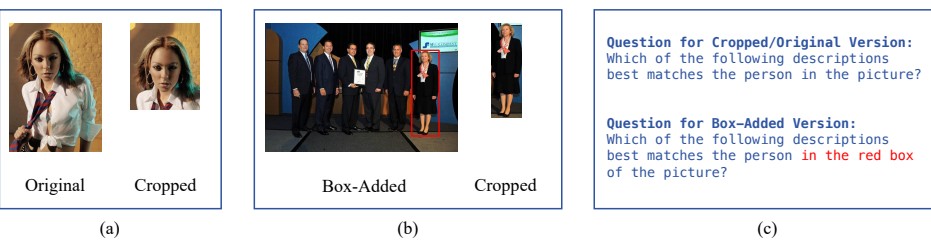

Figure 4: (a) The versions used for the three face understanding abilities. (b) The versions used for human attribute recognition. (c) When MLLMs are evaluated with different versions, the wording of the questions varies slightly.

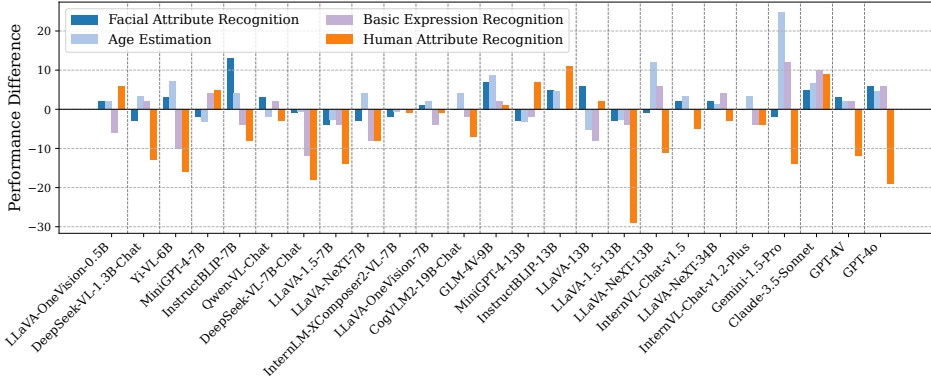

Figure 5: The performance differences between the two versions across various models. For the three face understanding abilities, we show the performance of the original version minus that of the cropped version. For human attribute recognition, we show the performance of the box-added version minus that of the cropped version.

We investigate the impact of the relative position of targets on performance in four L3 abilities: facial attribute recognition, age estimation, basic expression recognition, and human attribute recognition. As shown in Figure 4, for the three face understanding abilities, we provide both the original and cropped versions, where only one person is included but the relative position varies. For human attribute recognition, we offer box-added and cropped versions. In the box-added version, multiple people are included, with the target to be discussed indicated by a red box. Figure 5 illustrates the performance differences between the two versions across various models. Our findings can be summarized as follows.

**Face Understanding Abilities.** (1) Preferences for either version depend on the model and the ability, with no overarching trend observed. (2) A model's preference can vary across different face understanding abilities. For example, Yi-VL-6B shows no significant preference for facial attribute recognition, prefers the original images for age estimation, and favors cropped images for basic expression recognition. We think that this phenomenon may occur because MLLMs have been trained using images with different target relative positions when aligning visual information for different facial features.

**Human Attribute Recognition.** The majority of models perform better on the cropped version. This indicates that these models still struggle to accurately understand a specific individual when there are multiple people in the image.

We define the relative position sensitivity score (RPSS) as the sum of the absolute differences in scores between the two versions across the four tasks. This metric can serve as an effective reference

Table 2: Scores of the best open-source model, InternVL-Chat-v1.2-Plus, and the best closed-source model, GPT-4o, under different settings. ZS is short for Zero-Shot, H is short for Hints, VCoT is short for Vanilla CoT, 1TCoT is short for 1-stage Task-specific CoT. 2TCoT is short for 2-stage Task-specific CoT. Q is short for Question. O is short for Options. A is short for Answer. R is short for Relevant Analysis. The highest scores for open-source and closed-source MLLMs are marked in blue and green respectively.

| Setting | Format | Open-Source: InternVL-Chat-v1.2-Plus | | | | | Close-Source: GPT-4o | | | | |
|---------|--------|------|-------|------|------|---------|------|-------|------|------|---------|
| | | Face | Human | Per. | Rea. | Overall | Face | Human | Per. | Rea. | Overall |
| ZS | QO→A | 69.7 | 83.1 | 76.7 | 76.0 | 76.4 | 68.5 | 71.6 | 68.9 | 71.7 | 70.0 |
| H | QOH→A | 68.4 | 83.2 | 76.4 | 75.9 | 75.9 | 72.2 | 74.6 | 70.4 | 78.0 | 73.4 |
| H+VCoT | QOH→RA | 69.1 | 82.5 | 75.9 | 74.8 | 75.7 | 76.4 | 80.7 | 78.2 | 77.2 | 78.6 |
| H+1TCoT | QOH→RA | 68.6 | 81.4 | 75.6 | 74.3 | 75.0 | 77.9 | 81.9 | 79.0 | 81.2 | 79.9 |
| H+2TCoT | QOH→R, QOHR→A | 69.1 | 79.1 | 75.8 | 71.8 | 74.1 | 77.0 | 81.2 | 78.4 | 77.2 | 79.1 |

for training MLLMs with more robust visual alignment for face and human understanding. We observe that InternLM-XComposer2-VL-7B, LLaVA-OneVision-7B, InternVL-Chat-v1.5, LLaVA-NeXT-34B, and InternVL-Chat-v1.2-Plus not only perform well in the four tasks but also exhibit low sensitivity scores. Among them, InternLM-XComposer2-VL-7B has the lowest sensitivity score of only 3.7%.[6]

## 3.5 CoT PROMPTING

In this section, we select InternVL-Chat-v1.2-Plus and GPT-4o to explore whether incorporating hints and Chain-of-Thought (CoT) instructions in the prompts can enhance the MLLMs' performance. These two models have achieved the best overall performance in the main experiment among open-source models and closed-source models respectively. A hint involves tips on how to answer the question. For example, the hint for person re-identification is "if two people have significant differences in posture and their faces are relatively blurry, the main basis for determining whether they are the same person is their clothing characteristics." CoT instructions, on the other hand, guide MLLMs to articulate the reasoning process leading to the answer. The vanilla CoT instruction simply requires the model to "analyze the question and options step by step", whereas task-specific

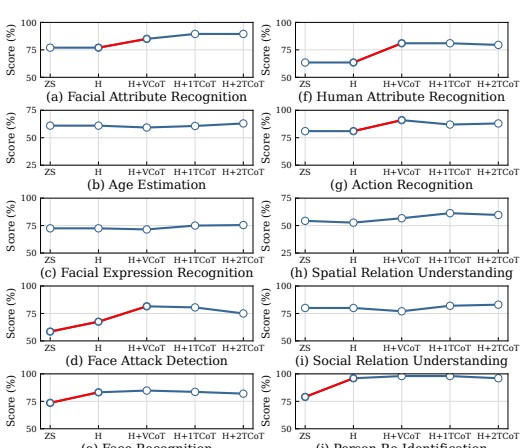

Figure 6: Main reasons of performance improvements for each L2 ability are highlighted in red.

CoT instructions provide more tailored guidance based on the task. For example, for the deepfake detection task, the prompt might instruct the model to "analyze whether there are any artifacts in the facial image." Following Multi-modal CoT (Zhang et al., 2024), we also conduct ablation experiments with both 1-stage and 2-stage frameworks. In the 1-stage framework, MLLMs are required to sequentially output the relevant analysis (rationale) and the answer in one round of dialogue. In the 2-stage framework, MLLMs first output the relevant analysis (rationale) in the first round and then provide the answer in the second round. Hints and task-specific CoT instructions for each L3 ability can be found in Appendix B.2.3.

Table 2 presents the performance of InternVL-Chat-v1.2-Plus and GPT-4o after incorporating hints and three different CoT settings. The results indicate that including hints and CoT instructions does not improve the performance of the open-source model; in fact, it may even cause a slight performance decline. By analyzing the outputs, we find that the open-source model does not provide rationales in its responses after adding CoT instructions to prompts. We believe this could be due to the model's insufficient generalization capabilities, preventing it from understanding the CoT instructions. In contrast, the closed-source GPT-4o shows significant performance improvements.

---

[6]For more models' RPSS, please refer to the Appendix C.4.

Table 3: Comparison between MLLMs and specialist models on 13 L3 abilities. The best-performing MLLMs are highlighted in blue, while abilities where MLLMs perform significantly worse than specialist models are marked in orange.

| L3 Ability | Age | Expression | | Deepfake | Spoofing | Action | Counting |
|---|---|---|---|---|---|---|---|
| Dataset | UTKFace | RAF-DB (Basic) | RAF-DB (Compound) | FF++ | SiW-Mv2 | HICO-DET | ShTech-A |
| Matric | MAE ↓ | ACC ↑ | ACC ↑ | ACC ↑ | ACER ↓ | mAP ↑ | MAE ↓ |
| Random | 27.89 | 13.85 | 8.08 | 50.84 | 50.05 | 9.32 | 1512.65 |
| InternVL-Chat-v1.5 | 6.43 | 72.23 | 42.93 | 56.21 | 14.84 | 22.29 | 2195.69 |
| LLaVA-NeXT-34B | 6.01 | 77.71 | 41.04 | 53.42 | 22.38 | 13.74 | 1592.55 |
| InternVL-Chat-v1.2-Plus | 5.21 | 76.40 | 30.56 | 52.89 | 19.92 | 12.25 | 2518.25 |
| Best of The Above 3 | 5.21 | 77.71 | 42.93 | 56.21 | 14.84 | 22.29 | 1592.55 |
| Early Specialist Model | 5.47 | 74.20 | 44.55 | 82.01 | 9.40 | 19.81 | 110.20 |
| Relative Score | 1.01 | 1.06 | 0.96 | 0.17 | 0.87 | 1.24 | -0.06 |
| Need Specialist Model | No. | No. | No. | Yes. | No. | No. | Yes. |

| L3 Ability | Basic FR | C.P. FR | C.A. FR | S.L. FR | Occ. FR | | Re-ID |
|---|---|---|---|---|---|---|---|
| Dataset | LFW | CPLFW | CALFW | SLLFW | MLFW | | Market1501 |
| Matric | ACC ↑ | ACC ↑ | ACC ↑ | ACC ↑ | ACC ↑ | | ACC ↑[7] |
| Random | 50.05 | 49.75 | 50.12 | 50.18 | 50.05 | | 49.47 |
| InternVL-Chat-v1.5 | 83.68 | 58.13 | 61.40 | 56.72 | 52.15 | | 77.53 |
| LLaVA-NeXT-34B | 91.32 | 65.87 | 62.07 | 70.25 | 53.73 | | 85.67 |
| InternVL-Chat-v1.2-Plus | 92.57 | 67.98 | 66.50 | 68.50 | 58.65 | | 88.73 |
| Best of The Above 3 | 92.57 | 67.98 | 66.50 | 70.25 | 58.65 | | 88.73 |
| Early Specialist Model | 99.50 | 87.47 | 92.43 | 98.40 | 82.87 | | 95.26 |
| Relative Score | 0.86 | 0.48 | 0.39 | 0.42 | 0.26 | | 0.86 |
| Need Specialist Model | No. | Yes. | Yes. | Yes. | Yes. | | No. |

Adding hints leads to a 3.4% improvement compared to the zero-shot setting. Building upon this, vanilla CoT, 1-stage task-specific CoT, and 2-stage task-specific CoT further improve performance by 5.2%, 6.5%, and 5.7%, respectively. Ultimately, the combination of hints and 1-stage task-specific CoT instructions emerge as the best setting for overall performance.

In Figure 6, we further explore the main reasons for the performance improvements of GPT-4o in each ability at L2. Hints significantly improve performance in face attack detection, face recognition, and person re-identification, while CoT instructions significantly improve performance in facial attribute recognition, face attack detection, human attribute recognition, and action recognition. For the reasons behind the performance improvements in each ability at L3, please refer to Appendix C.5.

## 3.6 SPECIALIST MODELS SIGNIFICANTLY OUTPERFORMING MLLMS

In this section, we explore whether specialist models corresponding to 13 L3 abilities can be used to enhance MLLMs. [8] We directly test the performance of MLLMs using original datasets from the face and human community to facilitate comparison with specialist models. We design a set of prompt templates to transform the classification problems into multiple-choice problems and the regression problems (age estimation and crowd counting) into fill-in-the-blank problems. [9] Specialist models are generally trained and tested on data from the same distribution. They can achieve high performance even if the test labels contain noise. However, the visual information learned by MLLMs and the original datasets used for testing may exhibit data distribution bias. To enable an effective comparison, we utilize early specialist models (which emerged after the widespread adoption of deep learning) as a reference to judge the performance of MLLMs on these tasks.[10]

We further define the relative performance score $S$ to normalize performances across different tasks: $S = (P_m - P_r)/(P_s - P_r)$, where $P_m$ is the performance of the MLLM. Here, we take the highest-performing model among InternVL-Chat-v1.2-Plus, LLaVA-Next-34B , and InternVL-Chat-v1.5 (the top three models in the main experiment). $P_r$ is the performance of random responses, and $P_s$ is the performance of the early specialist model. This metric typically ranges from 0 to 1, where a higher relative score indicates stronger abilities of MLLMs on the corresponding task. A relative

---

[7] The original metric for Market1501 is mAP. For easier comparison, we create a new testing protocol consisting of 750 positive pairs and 750 negative pairs. The ACC can be calculated in the same way as for LFW. We re-evaluate the early specialist model for Re-ID using the new protocol.

[8] We explain the reasons for not conducting experiments on the remaining 5 L3 abilities in Appendix B.3.1.

[9] For prompt templates, please refer to Appendix B.3.2.

[10] For the early specialist models used for comparison, please refer to Appendix C.6.

score below 0 stands for even worse results than random responses, whereas a score above 1 indicates the performance surpassing the corresponding specialist models for reference. As shown in Table 3, there is no need for MLLMs to introduce specialist models to enhance the response quality when the abilities of age estimation, facial expression recognition, face anti-spoofing, action recognition, and person re-identification are needed. In contrast, for deepfake detection and crowd counting tasks, the MLLM significantly underperforms specialist models. Moreover, for face recognition, MLLMs can approach the specialist model under the basic scenario but indicate poor performance under more challenging scenarios, such as cross-pose, cross-age, similar-looking, and occluded. To sum up, we recommend incorporating the corresponding specialist models into multi-modal assistants for applications where deepfake detection, crowd counting, and accurate face recognition are required. Appendix F provides a demonstration of how to enhance multi-modal assistant responses with specialist models.

## 4 RELATED WORK

**Evaluation of MLLMs about Face and Human Understanding.** Currently, there is no dedicated benchmark evaluating the face and human understanding abilities of MLLMs. Some efforts aim at comprehensively benchmarking MLLMs, containing some ability dimensions about face and human understanding. LAMM (Yin et al., 2023) evaluates 9 different 2D vision tasks using 11 existing public datasets. Among these, the facial classification task utilizes the CelebA (Liu et al., 2015) dataset to evaluate the accuracy of smile detection and hair attribute classification. MME (Fu et al., 2023) includes the celebrity recognition ability, requiring MLLMs to respond with Yes/No answers. SEED-Bench (Li et al., 2023a) includes the action recognition ability, where the inputs consist of multiple frames taken from a video, and MLLMs are required to choose the correct answer from four descriptions. MMBench (Liu et al., 2023c) includes the most extensive set of abilities related to faces and humans: celebrity recognition, action recognition, identity reasoning, and social relation, all of which are tested using multiple-choice problems. Considering the importance of faces and humans in multimedia, these evaluations are insufficient.

**Face and Human Understanding.** Face and human understanding is among the earliest research topics in artificial intelligence with successful applications. During the 2010s, the introduction of deep learning, particularly convolutional neural networks, significantly advanced face and human perception. In that era, numerous high-quality datasets were proposed for training and evaluating tasks of face attribute recognition (Liu et al., 2015), age estimation (Rothe et al., 2015; Escalera et al., 2015; Zhang et al., 2017), facial expression recognition (Barsoum et al., 2016; Li et al., 2017b; Mollahosseini et al., 2019), deepfake detection (Rössler et al., 2019; Dolhansky et al., 2019), face anti-spoofing (Liu et al., 2018; 2019), face recognition (Yi et al., 2014; Guo et al., 2016; Zheng et al., 2017; Deng et al., 2017; Zheng & Deng, 2018), human attribute recognition (Li et al., 2016; Liu et al., 2017), human-object interaction detection (Gupta & Malik, 2015; Xu et al., 2019), crowd counting (Zhang et al., 2016), social relationship recognition Sun et al. (2017); Li et al. (2017a) and person re-ideitification Li et al. (2014); Zheng et al. (2015). Entering the 2020s, a new paradigm emerged, which initially pre-trains a task-agnostic backbone and then based on this, trains a unified face or human model (Ci et al., 2023; Wang et al., 2023b; Qin et al., 2024) to simultaneously handle multiple face and human understanding tasks within a unified structure. In our evaluation, we observe that in certain tasks, MLLMs do not perform as well as specialist models. Utilizing these unified face or human models as the specialist models to help MLLMs can greatly facilitate deployment.

## 5 CONCLUSION

In this work, we propose the hierarchical Face-Human-Bench, the first benchmark specifically designed to evaluate MLLMs' face and human understanding abilities. We comprehensively and scientifically assess the performance of 25 mainstream MLLMs with our benchmark. We reveal the correlations between abilities and explore the impact of the relative position of targets and CoT prompting on the performance of MLLMs. Inspired by multimodal agents, we investigate which abilities of MLLMs need to be supplemented by specialist models. Our work will provide the face and human community valuable insights on how to more effectively leverage multi-modal assistants in applications related to "faces and humans."

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

# Appendix

CONTENTS

## A MORE DETAILS ON FACE-HUMAN-BENCH

### A.1 DEFINITION ABOUT EACH LEAF ABILITY

We will sequentially describe the definitions of L2 abilities and the L3 abilities they encompass. We provide examples of problems in Face-Human-Bench in Tables 4 to 11.

**Facial Attribute Recognition**: Recognize various characteristics and traits from facial images.

**Age Estimation**: Estimate the age of the person in the image based on facial information.

**Facial Expression Recognition**: Recognize the emotions of the person in the image, categorized into basic and compound types. Basic expressions include surprised, fearful, disgusted, happy, sad, angry, and neutral. Compound expressions provide more nuanced emotional descriptions, including: happily surprised, happily disgusted, sadly fearful, sadly angry, sadly surprised, sadly disgusted, fearfully angry, fearfully surprised, angrily surprised, angrily disgusted, and disgustedly surprised.

**Face Attack Detection**: Determine whether the face in the image involves digital manipulation or physical spoofing. The corresponding sub-abilities are referred to as Deepfake Detection and Face Anti-Spoofing, respectively.

**Face Recognition** Identify and verify individuals' identities in images according to facial information. In our tests, this ability is mainly to determine whether two photos showcase the same individual. Five scenarios are involved: basic, cross-pose, cross-age, similar-looking, and occluded.

**Human Attribute Recognition** Recognize various characteristics and traits from human images.

**Action Recognition** Recognize human actions, including interactions with objects.

**Spatial Relation Understanding** Understand the spatial positions of people in the image, including relative position understanding (comprehending the relative positions of one person to others and objects) and crowd counting (counting the number of people in the image).

**Social Relation Understanding** Including social relationship recognition (inferring social relationships between people through their interactions) and identity reasoning (deducing social identity based on a person's attributes, actions, interactions with others, and environmental information).

**Person Re-Identification** Identify and verify individuals' identities in images based on full-body attributes (usually excluding the face, as facial features are often blurry).

Table 4: Examples of problems in Face-Human-Bench.

| Ability | Example |
|---|---|
| Facial Attribute Recognition | **Image:**  
 **Question:** 
 Please select the description that best applies to the person in the picture. 
 A. not wearing necktie, not wearing lipstick, not wearing earrings. 
 B. without eyeglasses, bald, with mouth slightly open. 
 C. male, with black hair, wearing earrings. 
 D. with eyeglasses, not wearing hat, with bangs. 
 **Answer:** A. |

Table 5: Examples of problems in Face-Human-Bench.

| Ability | Example |
| --- | --- |
| Age Estimation (5-Year Interval) | **Image:** 
**Question:**
Which age do you believe is most likely for the person in the photo?
A. 10. B. 15. C. 20. D. 25.
**Answer:** D. |
| Age Estimation (10-Year Interval) | **Image:** 
**Question:**
Which of the following ages is the most likely for the person in the picture?
A. 20. B. 30. C. 40. D. 50.
**Answer:** A. |
| Age Estimation (15-Year Interval) | **Image:** 
**Question:**
Which of the following ages is the most likely for the person in the picture?
A. 47. B. 62. C. 77. D. 92.
**Answer:** B. |
| Facial Expression Recognition (Basic Expression Recognition) | **Image:** 
**Question:**
What is the expression of the person in this photo?
A. Neutral.
B. Sadness.
C. Disgust.
D. Fear.
**Answer:** A. |
| Facial Expression Recognition (Compound Expression Recognition) | **Image:** 
**Question:**
Based on this picture, what is the person's expression?
A. Happily Disgusted.
B. Fearfully Surprised.
C. Sadly Disgusted.
D. Sadly Fearful.
**Answer:** A. |

Table 6: Examples of problems in Face-Human-Bench.

| Ability | Example |
|---|---|
| Face Attack Detection (Deepfake Detection) (Bona Fide) | **Image:**  **Question:** Has the facial image undergone digital alteration? A. yes. B. no. **Answer:** B. |
| Face Attack Detection (Deepfake Detection) (Face-Swapping) | **Image:**  **Question:** Was the facial image digitally modified in any way? A. yes. B. no. **Answer:** A. |
| Face Attack Detection (Deepfake Detection) (Face-Reenactment) | **Image:**  **Question:** Was the facial appearance digitally changed? A. yes. B. no. **Answer:** A. |
| Face Attack Detection (Face Anti-Spoofing) (Bona Fide) | **Image:**  **Question:** Has the facial image been compromised by a presentation attack? A. yes. B. no. **Answer:** B. |
| Face Attack Detection (Face Anti-Spoofing) (Print) | **Image:**  **Question:** Is there a spoofing attempt visible in the facial image? A. yes. B. no. **Answer:** A. |
| Face Attack Detection (Face Anti-Spoofing) (Replay) | **Image:**  **Question:** Is the facial recognition being deceived by a presentation attack? A. yes. B. no. **Answer:** A. |

Table 7: Examples of problems in Face-Human-Bench.

| Ability | Example |
|---|---|
| Face Recognition (Basic Face Recognition) | **Image:**  **Question:** Are the people portrayed in the two pictures identical? A. yes. B. no. **Answer:** A. |
| Face Recognition (Basic Face Recognition) | **Image:**  **Question:** Are the individuals in both images one and the same? A. yes. B. no. **Answer:** B. |
| Face Recognition (Cross-Pose Face Recognition) | **Image:**  **Question:** Do the individuals appearing in the two images happen to be identical? A. yes. B. no. **Answer:** A. |
| Face Recognition (Cross-Pose Face Recognition) | **Image:**  **Question:** Do the people shown in both pictures happen to be one and the same person? A. yes. B. no. **Answer:** B. |
| Face Recognition (Cross-Age Face Recognition) | **Image:**  **Question:** Are the people portrayed in the two pictures identical? A. yes. B. no. **Answer:** A. |

Table 8: Examples of problems in Face-Human-Bench.

| Ability | Example |
|---------|---------|
| Face Recognition (Cross-Age Face Recognition) | **Image:**  **Question:** Do the individuals in both images happen to be the same person? A. yes. B. no. **Answer:** B. |
| Face Recognition (Similar-Looking Face Recognition) | **Image:**  **Question:** Are the persons depicted in the photos on the left and right sides identical? A. yes. B. no. **Answer:** A. |
| Face Recognition (Similar-Looking Face Recognition) | **Image:**  **Question:** Are the persons depicted in the photos on the left and right sides identical? A. yes. B. no. **Answer:** B. |
| Face Recognition (Occluded Face Recognition) | **Image:**  **Question:** Is the individual captured in both the left and right photographs one and the same person? A. yes. B. no. **Answer:** A. |
| Face Recognition (Occluded Face Recognition) | **Image:**  **Question:** Do the individuals appearing in the two images happen to be identical? A. yes. B. no. **Answer:** B. |

Table 9: Examples of problems in Face-Human-Bench.

| Ability | Example |
|---|---|
| Human Attribute Recognition | **Image:** 
**Question:**
Which statement best describes the individual highlighted in the red box in the picture?
A. She is wearing a long-sleeve shirt and is not wearing a hat or a skirt.
B. She is wearing a T-shirt and a hat, but her clothes do not have any logos.
C. She is dressed informally in a skirt and wearing sunglasses.
D. She has long hair and is wearing a short-sleeved shirt along with a face mask.
**Answer:** A. |
| Action Recognition | **Image:** 
**Question:**
Which of these options best describes what the person in the red box is doing in the picture?
A. Washing the motorcycle.
B. Waxing the motorcycle.
C. Polishing the motorcycle.
D. Repairing the motorcycle.
**Answer:** A. |
| Spatial Relation Understanding (Relative Position Understanding) | **Image:** 
**Question:**
Among the following options, what is the most fitting way to characterize the subject (marked with a red box)'s location in relation to the object (marked with a green box)?
A. The child is behind the sofa.
B. The child is to the right of the sofa.
C. The child is to the left of the sofa.
D. The child is under the sofa.
**Answer:** A. |

Table 10: Examples of problems in Face-Human-Bench.

| Ability | Example |
|---------|---------|
| Spatial Relation Understanding (Crowd Counting) (Less than 10) | **Image:**  **Question:** What's the number of individuals in this picture? A. 2. B. 3. C. 4. D. 5. **Image:** D. |
| Spatial Relation Understanding (Crowd Counting) (10-100) | **Image:**  **Question:** Among the options, which numeral is closest to the total count of humans in the picture? A. 10. B. 30. C. 90. D. 140. **Image:** B. |
| Spatial Relation Understanding (Crowd Counting) (More than 100) | **Image:**  **Question:** What is the closest numerical value among the options to the number of individuals in the image? A. 400. B. 1100. C. 3200. D. 5300. **Answer:** B. |
| Social Relation Understanding (Social Relationship Recognition) | **Image:**  **Question:** Which relationship do the two people in the red box in the photo most likely have? A. Couple. B. No Relation. C. Family. D. Friends. **Answer:** A. |
| Social Relation Understanding (Identity Reasoning) | **Image:**  What is the most likely occupation of the person highlighted in red in the picture? A. basketball player. B. basketball team manager. C. basketball coach. D. sports commentator. **Answer:** A. |

Table 11: Examples of problems in Face-Human-Bench.

| Ability | Example |
|---------|---------|
| Person Re-Identification | **Image:** 

 **Question:** 
 Is the person in the first picture the same as the person in the second picture? 
 A. yes. 
 B. no. 
 **Answer:** A. |
| Person Re-Identification | **Image:** 

 Is the individual captured in both the left and right photographs one and the same person? 
 A. yes. 
 B. no. 
 **Image:** B. |

## A.2 DATA SOURCES AND STATISTICS

Table 12 provides information on the data sources for Face-Human-Bench, as well as the image processing pipeline, the number of problems in the development and test sets, and the weights, for each subset.

We set the weights of all 10 L2 abilities to be equal. For L2 abilities that encompass multiple L3 abilities, each L3 ability equally shares the weight of the corresponding L2 ability. For L3 abilities that encompass multiple image versions, each image version subset equally shares the weight of the corresponding L3 ability. Finally, we obtain the detailed weights of each subset, as shown in Table 12.

We sequentially provide overviews of the public datasets we used for original samples.

**CelebA** (Liu et al., 2015) is a large-scale facial attributes dataset released by the Multimedia Laboratory of Chinese University of Hong Kong. It contains over 200,000 celebrity images, each annotated with 40 attributes. The dataset includes a wide range of body pose variations and complex, diverse background information. It comprises 10,177 identities, 202,599 face images, and 5 landmark positions, with 40 binary attribute annotations for each image.

**UTKFace** (Zhang et al., 2017) dataset is a large-scale facial dataset with a wide age range, spanning from 0 to 116 years. It contains over 20,000 face images, annotated with age, gender, and ethnicity labels.

**RAF-DB** (Li et al., 2017b) is a large-scale facial expression database consisting of 29,672 real-world images, each accompanied by a 7-dimensional expression distribution vector. It includes two different subsets: a single-label subset with 7 basic expressions (RAF-DB Basic) and a two-tab subset with 12 compound expressions (RAF-DB Compound). Additionally, the dataset provides 5 precise landmark locations, 37 automatic landmark positions, bounding boxes, and annotations for ethnicity, age range, and gender attributes for each image.

**FF++** (Rössler et al., 2019) consists of 1,000 original video sequences processed using four different automated facial manipulation methods: Deepfakes, Face2Face, FaceSwap, and NeuralTextures. The data in FaceForensics++ comes from 977 YouTube videos, all featuring trackable frontal faces without occlusions, allowing the automated manipulation methods to generate realistic forgeries.

Table 12: Data sources and statistics of the Face-Human-Bench.

| Level-1 | Level-2 | Level-3 | Data Source | $p_{image}$ | Dev. Num. | Test Num. | Weight |
|---|---|---|---|---|---|---|---|
| Face | Facial Attribute Recognition | Facial Attribute Recognition | CelebA | Identity | 50 | 100 | 5.0% |
| | | | | Crop | 50 | 100 | 5.0% |
| | Age Estimation | Age Estimation | UTKFace | Identity | 75 | 150 | 5.0% |
| | | | | Crop | 75 | 150 | 5.0% |
| | Facial Expression Recognition | Basic Expression Recognition | RAF-DB (Basic) | Identity | 25 | 50 | 2.5% |
| | | | | Crop | 25 | 50 | 2.5% |
| | | Compound Expression Recognition | RAF-DB (Compound) | Identity | 25 | 50 | 5.0% |
| | Face Attack Detection | Deepfake Detection | FF++ | Identity | 50 | 100 | 5.0% |
| | | Face Anti-Spoofing | SiW-Mv2 | Identity | 50 | 100 | 5.0% |
| | Face Recognition | Basic Face Recognition | LFW | Cat | 25 | 50 | 2.0% |
| | | Cross-Pose Face Recognition | CPLFW | Cat | 25 | 50 | 2.0% |
| | | Cross-Age Face Recognition | CALFW | Cat | 25 | 50 | 2.0% |
| | | Similar-Looking Face Recognition | SLLFW | Cat | 25 | 50 | 2.0% |
| | | Occluded Face Recognition | MLFW | Cat | 25 | 50 | 2.0% |
| Human | Human Attribute Recognition | Human Attribute Recognition | WIDER Attribute | AddBox | 50 | 100 | 5.0% |
| | | | | Crop | 50 | 100 | 5.0% |
| | Action Recognition | Action Recognition | HICO-DET | AddBox | 50 | 100 | 10.0% |
| | Spatial Relation Understanding | Relative Position Understanding | SpatialSense | Identity | 25 | 50 | 5.0% |
| | | Crowd Counting | PISC ShTech | Identity | 75 | 150 | 5.0% |
| | Social Relation Understanding | Social Relationship Recognition | PISC | AddBox | 25 | 50 | 5.0% |
| | | Identity Reasoning | PISC | AddBox | 25 | 50 | 5.0% |
| | Person Re-Identification | Person Re-Identification | Market-1501 | Cat | 50 | 100 | 10.0% |

**SiW-Mv2** (Guo et al., 2022) collects 785 videos from 493 subjects, and 915 spoof videos from 600 subjects. The dataset includes 14 types of spoofing, ranging from typical print and replay attack, to various masks, impersonation makeup and physical material coverings. SiW-Mv2 exhibits a good variance in spoofing modes, with each mode specified and validated by the IARPA project.

**LFW** (Huang et al., 2008) is a commonly used test set for face recognition, comprising 13,233 face images sourced from natural scenes in everyday life. Each image is associated with a name, representing 5,749 individuals, with most people having only one image. The database randomly selected 6,000 pairs of faces to create face recognition image pairs to test the accuracy of face recognition systems, with 3,000 pairs containing two images of the same person and 3,000 pairs featuring one image of different individuals.

**CPLFW** (Zheng & Deng, 2018) builds upon LFW by considering the impact of pose variations. It specifically searches for and selects 3,000 pairs of positive faces with differing poses, adding pose variation to the intra-class variance. Additionally, it includes negative pairs with the same gender and ethnicity to minimize the influence of attribute differences between positive and negative pairs.

**CALFW** (Zheng et al., 2017) builds upon LFW by considering the impact of age variations. It specifically searches for and selects 3,000 pairs of positive faces with age differences to increase the intra-class variance associated with the aging process. Negative pairs are chosen to have the same gender and ethnicity to reduce the influence of attribute differences.

**SLLFW** (Deng et al., 2017) intentionally selects 3,000 pairs of visually similar faces through human crowdsourcing from the original image folder, replacing the random negative pairs in LFW.

**MLFW** (Wang et al., 2022) dataset is created based on CALFW and focuses on masked faces. The masks generated for the faces in the dataset maintain good visual consistency with the original faces. It includes a variety of mask templates that cover most common styles encountered in everyday life, achieving diversity of the samples.

**WIDER Attribute** (Li et al., 2016) is a large-scale human attributes dataset containing 13,789 images across 30 scene categories, with 57,524 human bounding boxes. Each bounding box is annotated with 14 binary attributes, including male, long hair, sunglasses, hat, long shirt, long sleeves, formal, shorts, jeans, long pants, skirt, mask, logo, and checkered or striped patterns.

**HICO-DET** Xu et al. (2019) is a commonly used dataset in the Human Object Interaction (HOI) domain, consisting of 47,776 images, with 38,118 in the training set and 9,658 in the testing set. The dataset includes 117 action (verb) categories, 80 object categories, and 600 verb-object combinations.

**SpatialSense** Yang et al. (2019) is a dataset for spatial relation recognition, where the task is to determine whether a specific spatial relation holds between two given objects. The dataset contains 17,498 relations on 11,569 images, involving 3,679 unique object classes, with 2,139 of these classes appearing only once, presenting a challenging long-tail distribution.

**PISC** Li et al. (2017a) is focused on the task of social relationship recognition in still images. It is used to benchmark models that analyze the relationships between people based on contextual and individual features. It contains 22,670 images with 76,568 annotated samples representing 9 types of social relationships.

**ShTech** Zhang et al. (2016) is focused on the task of crowd counting, where the goal is to accurately estimate the number of people in an image with varying crowd density and perspective. It contains 1,198 images with approximately 330,000 annotated heads. The dataset aims to address challenges in crowd counting that were not covered by previous datasets.

**Market-1501** Zheng et al. (2015) is designed for the task of person re-identification. This dataset addresses the limitations of scale and realistic conditions found in previous datasets. The large-scale data supports training and testing models effectively for person re-identification. It includes over 32,000 annotated bounding boxes and a distractor set of more than 500,000 images.

### A.3 More Details on the Semi-Automatic Data Pipeline

### A.3.1 Details on Image Processing Pipeline

Figure 7 illustrates four operations of the image processing pipeline: cropping, concatenating, adding boxes, or leaving the original images unchanged. For simplicity, these four operations are denoted as Crop, Cat, AddBox, and Identity, respectively. The image processing pipeline used for each L3 ability is shown in Table 12.

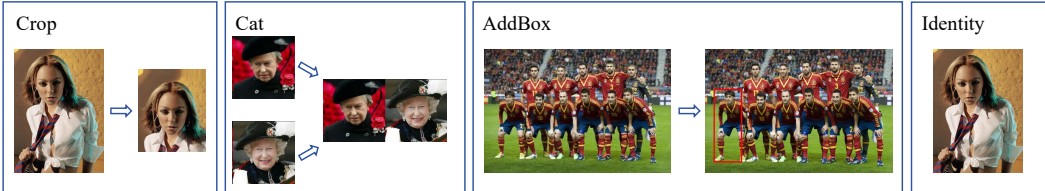

Figure 7: Four operations of the image processing pipeline.

### A.3.2 Details on Text Processing Pipeline

We introduce the text processing pipeline for each L3 ability as follows. **Facial Attribute Recognition** Each option involves three attributes. At least two of the three attribute descriptions are incorrect in the incorrect options.

**Age Estimation** Add incorrect options at intervals of 5 years, 10 years, and 15 years, with each interval accounting for one-third of the total.

**Basic Expression Recognition** Incorrect options are randomly selected from the remaining 6 categories of expressions after removing the correct option.

**Compound Expression Recognition** Incorrect options are randomly selected from the remaining 10 categories of expressions after removing the correct option.

**Deepfake Detection** Set the options to "Yes" and "No". "Yes" indicates the presence of digital manipulations, while "No" indicates their absence.

**Face Anti-Spoofing** Set the options to "Yes" and "No". "Yes" indicates the presence of physical spoofs, while "No" indicates their absence.

**Basic/Cross-Pose/Cross-Age/Similar-Looking/Occluded Face Recognition** Set the options to "Yes" and "No". "Yes" indicates that the two photos are of the same person, while "No" indicates that the two photos are not of the same person.

**Human Attribute Recognition** Each option involves three attributes combined into a complete sentence using ChatGPT. At least two of the three attribute descriptions are incorrect in the incorrect options.

**Action Recognition** The incorrect options are actions generated by ChatGPT related to but not the same as the correct option.

**Relative Position Understanding** Each option is a sentence formed by connecting the subject and the object with a preposition. Incorrect options are generated by randomly selecting prepositions from the remaining 8 categories of relative positions after removing the correct preposition.

**Crowd Counting** The set includes three equally sized subsets, with the number of people in each subset being within the ranges of less than 10, 10-100, and more than 100, respectively. In the first subset, the incorrect options are also numbers within 10. In the latter two subsets, the incorrect options are numbers that are half, three times, and five times the correct option, respectively, with all options rounded to the nearest 10 and 100.

**Social Relationship Recognition** Incorrect options are randomly selected from the remaining 5 categories of social relations after removing the correct option.

**Identity Reasoning** The incorrect options are occupations generated by GPT related to but not the same as the correct option.

**Person Re-Identification** Set the options to "Yes" and "No". "Yes" indicates that the two photos are of the same person, while "No" indicates that the two photos are not of the same person.

## B    MORE DETAILS ON EXPERIMENT SETUP

### B.1    OVERVIEWS OF INVOLVED MLLMS

**GPT-4V and GPT-4o**: GPT-4V (OpenAI, 2023b), released by OpenAI in September 2023, is a vision-enabled variant of the GPT-4 model, utilizing the same training process as GPT-4 for its visual capabilities. It is first trained on a large dataset of text and images, followed by fine-tuning through Reinforcement Learning with Human Feedback (RLHF). GPT-4V demonstrates the exceptional performance of a language-only system augmented with new modalities. The API we applied in our experiments is "gpt-4-turbo-2024-04-09". GPT-4o OpenAI (2024) is released by OpenAI in May 2024. It accepts any combination of text, image, audio and video as input and generates any combination of text, image, and audio output. GPT-4o attains GPT-4 Turbo-level performance in text, inference, and code, while also demonstrating strong capabilities in multilingual, audio, and visual tasks. The API we applied in our experiments is "gpt-4o-2024-05-13".

**Gemini** (Team et al., 2023): Gemini is a multimodal large model developed by Google, available in three scales: Ultra, Pro, and Nano. From its inception, Gemini was designed with a multimodal focus, excelling in tasks across image, audio, video, and text domains. In February 2024, Google released Gemini 1.5 (Reid et al., 2024), which includes Gemini 1.5 Pro and the more lightweight Gemini 1.5 Flash. In our work, we employ Gemini 1.5 Pro to conduct experiments.

**Claude** (Anthropic, 2023): The Claude model is developed by Anthropic and is intended to be a useful, honest and harmless assistant. The version we applied in this paper, Claude 3.5 Sonnet (Anthropic, 2024a), was released on June 2024. It is the most powerful visual model in the Claude series to date.

**LLaVA** (Liu et al., 2023b): LLaVA is an open-source large multimodal model that leverages multimodal language-image instruction-following data for instruction tuning. It was released in April 2023. LLaVA-1.5 (Liu et al., 2023a), released in October 2023, introduced the following key improvements: the use of MLP as a vision-language connector, the use of prompt data with explicitly specified output formats, and the addition of task-specific datasets for training. Following that, LLaVA-1.6 (LLaVA-NeXT) (Liu et al., 2024a) was released in January 2024, featuring improved input image resolution and enhanced visual reasoning and OCR capabilities. The model also supports better visual conversation on different scenarios and applications. SGLang was utilized for efficient deployment and inference. We apply LLaVA-13B, LLaVA-1.5-7B, LLaVA-1.5-13B, LLaVA-NeXT-7B, LLaVA-NeXT-13B, and LLaVA-NeXT-34B in our experiments.

**MiniGPT-4** (Zhu et al., 2024): MiniGPT-4, released in April 2023, uses a projection layer to align a frozen vision encoder with the frozen LLM Vicuna. The authors trained MiniGPT-4 in two stages: the first stage involved using a low-level dataset, and in the second stage, they curated a detailed image description dataset to fine-tune the model. In our experiments, we use MiniGPT-4-7B and MiniGPT-4-13B.

**InstructBLIP** (Dai et al., 2023): InstructBLIP, released in May 2023, applies its instruction-tuning paradigm to the BLIP-2 (Li et al., 2023b) model. To be specific, InstructBLIP performs instruction fine-tuning on visual tasks to enhance model performance. In our experiments, InstructBLIP-7B and InstructBLIP-13B are used.

**Qwen-VL** (Bai et al., 2023): Qwen-VL, released in August 2023, accepts images, text, and bounding boxes as inputs, and outputs text and bounding boxes. It supports multilingual and multi-image interleaved dialogue, as well as open-domain localization in Chinese. Qwen-VL is also capable of relatively fine-grained recognition and understanding. We adapt Qwen-VL-Chat in our experiments.

**InternLM-XComposer2-VL** (Zhang et al., 2023): InternLM-XComposer-VL, released in September 2023, is a multimodal large language model built with InternLM (Team, 2023) as the language model. Later, in January 2024, InternLM-XComposer2-VL (Dong et al., 2024) was released, supporting free-form text and image composition. The authors proposed the Partial LoRA (PLoRA) method, which balances precise visual understanding with literary-inspired text generation. InternLM-XComposer2-VL-7B is used in our experiments.

**Yi-VL** (Young et al., 2024): Yi-VL, released in May 2024, excels in image-text understanding and chat generation, supporting multi-turn image-text conversations, bilingual text, and fine-grained

image comprehension. Yi-VL adopts the LLaVA architecture and employs a three-stage training process to align visual information with the semantic space of Yi LLM (Young et al., 2024).

**InternVL** (Chen et al., 2023): InternVL, released in December 2023, extends its visual model to 6 billion parameters. It progressively aligns with the LLM using web-scale image-text data. InternVL-Chat-V1.2 was released in February 2024, expanding the LLM to 34 billion parameters. Shortly after, InternVL-Chat-v1.2-Plus was introduced, utilizing more supervised fine-tuning (SFT) data to further enhance its performance. Subsequently, InternVL-Chat-v1.5 (Chen et al., 2024) was released in April 2024, with improvements primarily focused on a stronger visual encoder, dynamic high-resolution capability, and a high-quality bilingual dataset. The model we use in the experiments includes InternVL-Chat-v1.2-Plus and InternVL-Chat-v1.5.

**DeepSeek-VL** (Lu et al., 2024): DeepSeek-VL, released in March 2024, is designed for general multimodal understanding. It is built for real-world applications in visual and language comprehension, capable of handling tasks such as logical diagrams, web pages, formula recognition, scientific literature, natural images, etc. In the experiments, we apply DeepSeek-VL-1.3B and DeepSeek-VL-7B.

**CogVLM2 and GLM-4V** (Wang et al., 2023a; Hong et al., 2024): CogVLM, released in October 2023, enables deep fusion of visual and language features without sacrificing performance on NLP tasks. In May 2024, the next generation, CogVLM2, was introduced. It inherited the visual expert architecture and improved training recipes in the pre-training and post-training stages, supporting high input resolutions. Shortly after, in June 2024, GLM-4V was released. It used the same data and training recipes as CogVLM2 but employed GLM-4-9B as the language models and removed the visual expert to reduce the model size. In our experiments, we utilize CogVLM2-19B-Chat and GLM-4V-9B.

**LLaVA-OneVision** (Li et al., 2024): LLaVA-OneVision, released in August 2024, supports three major computer vision scenarios: single image, multi-image, and video scenes. It also exhibits strong transfer learning capabilities across different modalities and scenarios. We use LLaVA-OneVision-0.5B and LLaVA-OneVision-7B in our experiments.

Table 13 summarizes the LLMs and vision encoders used in involved MLLMs.

Table 13: The LLMs and vision encoders used in involved MLLMs.

| Model | LLM | Params. | Vision Encoder | Params. |
|---|---|---|---|---|
| LLaVA-OneVision-0.5B | Qwen2-0.5B | 0.5B | SigLIP ViT-L/16 | 400M |
| DeepSeek-VL-1.3B-Chat | DeepSeek-LLM-1.3B-Base | 1.3B | SigLIP ViT-L/16 | 400M |
| Yi-VL-6B | Yi-6B | 6B | CLIP ViT-H/14 | 632M |
| MiniGPT-4-7B | Vicuna-7B | 7B | EVA-CLIP-g/14 | 1.0B |
| InstructBLIP-7B | Vicunad-7B | 7B | EVA-CLIP-g/14 | 1.0B |
| Qwen-VL-Chat | Qwen-7B | 7B | Open CLIP-G/14 | 1.8B |
| DeepSeek-VL-7B-Chat | DeepSeek-LLM-7B-Base | 7B | SigLIP ViT-L/16 + SAM ViT-B | 400M + 86M |
| LLaVA-1.5-7B | Vicuna-v1.5-7B | 7B | CLIP-L/14 | 304M |
| LLaVA-NeXT-7B | Vicuna-v1.5-7B | 7B | CLIP-L/14 | 304M |
| InternLM-XComposer2-VL-7B | InternLM-7B | 7B | EVA-CLIP-g/14 | 1.0B |
| LLaVA-OneVision-7B | Qwen2-7B | 7B | SigLIP ViT-L/16 | 400M |
| CogVLM2-19B-Chat | Llama-3-8B-Instruct | 8B | EVA-02-CLIP-E/14 | 4.4B |
| GLM-4V-9B | GLM-4-9B | 9B | EVA-02-CLIP-E/14 | 4.4B |
| MiniGPT-4-13B | Vicuna-13B | 13B | EVA-CLIP-g/14 | 1.0B |
| InstructBLIP-13B | Vicuna-13B | 13B | EVA-CLIP-g/14 | 1.0B |
| LLaVA-13B | LLaMA-2-13B-Chat | 13B | CLIP-L/14 | 304M |
| LLaVA-1.5-13B | Vicuna-v1.5-13B | 13B | CLIP-L/14 | 304M |
| LLaVA-NeXT-13B | Vicuna-v1.5-13B | 13B | CLIP-L/14 | 304M |
| InternVL-Chat-v1.5 | InternLM2-20B | 20B | InternViT-6B | 6B |
| LLaVA-NeXT-34B | Yi-34B | 34B | CLIP-L/14 | 304M |
| InternVL-Chat-v1.2-Plus | Nous-Hermes-2-Yi-34B | 34B | InternViT-6B | 6B |

## B.2 MORE DETAILS ON THE EXPERIMENTS FOR RQ1

### B.2.1 PROMPT TEMPLATES FOR DIFFERENT SETTINGS

**Zero-Shot (ZS)** The prompt template used for the zero-shot setting is shown in Table 14.

**Hints (H)** The prompt template for experiments with hints is shown in Table 15.

Table 14: The prompt template used for the zero-shot setting.

> Question: [Question]
> [Options]
> Please provide the answer to the multiple-choice question, using only the option's letter to indicate your choice. Note: Only one option is correct. For questions you are unsure about, please choose the answer you think is most likely.

Table 15: The prompt template used for experiments with hints.

> Question: [Question]
> [Options]
> Hint: [Hint]
> Please provide the answer to the multiple-choice question **based on the hint**, using only the option's letter to indicate your choice. Note: Only one option is correct. For questions you are unsure about, please choose the answer you think is most likely.

**Hints and Vanilla CoT Instructions (H+VCoT)** The prompt template for experiments with hints and vanilla CoT instructions is shown in Table 16.

Table 16: The prompt template used for experiments with hints and vanilla CoT instructions.

> Question: [Question]
> [Options]
> Hint: [Hint]
> First, please analyze the question and options **step by step** in conjunction with the input image.
> Then, please provide the answer to the multiple-choice question **based on the hint and relevant analysis**. Note: Only one option is correct. For questions you are unsure about, please choose the answer you think is most likely.

Table 17: The prompt template used for one-stage experiments with hints and task-specific CoT instructions.

> Question: [Question]
> [Options]
> Hint: [Hint]
> First, [Task-specific CoT instruction]
> Then, please provide the answer to the multiple-choice question **based on the hint and relevant analysis**. Note: Only one option is correct. For questions you are unsure about, please choose the answer you think is most likely.

**Hints and Task-Specific Instructions With One-Stage Framework (H+1TCoT)** The prompt template for one-stage experiments with hints and task-specific CoT instructions is shown in Table 17.

**Hints and Task-Specific Instructions With Two-Stage Framework (H+2TCoT)** The prompt template for two-stage experiments with hints and task-specific CoT instructions is shown in Table 18.

B.2.2   PROMPT USED FOR CHOICE EXTRACTION

The prompt used for choice extraction is shown in Table 19.

B.2.3   HINTS AND TASK-SPECIFIC COT INSTRUCTIONS

Hints and task-specific CoT instructions for each L3 ability are shown in Table 20.

Table 18: The prompt template used for two-stage experiments with hints and task-specific CoT instructions.

---

**Stage 1**
Question: [Question]
[Options]
Hint: [Hint]
[Task-specific CoT instruction]
**Stage 2**
Question: [Question]
[Options]
Hint: [Hint]
Relevant Analysis: [Output from stage 1]
Please provide the answer to the multiple-choice question **based on the hint and relevant analysis**.
Note: Only one option is correct. For questions you are unsure about, please choose the answer you think is most likely.

---

Table 19: The prompt template used for choice extraction.

---

You are an AI assistant to help me match an answer with several options of a multiple-choice problem. You are provided with a question, several options, and an answer, and you need to find which option is most similar to the answer. If the meaning of all options is significantly different from the answer, output X. You should output a single uppercase character in A, B, C, D (if they are valid options), and X.

Question: Please select the description that best matches the individual depicted.
Options:
A. He is wearing a face mask but is not wearing a hat or a skirt.
B. He is wearing a face mask, a hat, and shorts.
C. He has short hair and is not wearing a face mask or a T-shirt.
D. He is not wearing clothes with a logo or stripes, and he isn't wearing sunglasses.
Answer: He is wearing a face mask, a hat, and shorts.
Your Output: B

Question: Which description best represents the person in the image?
Options:
A. She is wearing a T-shirt and sunglasses, and her clothes do not have a logo.
B. She is wearing a face mask and sunglasses but is not wearing long pants.
C. She is without sunglasses, not wearing a hat, and not wearing a T-shirt.
D. She is dressed informally in a short-sleeved top and is not wearing a T-shirt.
Answer: None of the provided descriptions accurately represent the person in the image.
Your Output: X

Question: [Question]
Options: [Options]
Answer: [Answer]
Your Output:

---

Table 20: Hints and task-specific CoT instructions.

| L3 Ability | Hint | Task-specific CoT instruction |
|---|---|---|
| F. Attr. | / | Please analyze whether the characteristics described in the multiple-choice options match the attributes of the face in the image, one by one. |

| L3 Ability | Hint | Task-specific CoT instruction |
|---|---|---|
| Age | / | Please (1) analyze the facial age characteristics of the person in the image and (2) provide a possible age number that you think is appropriate. Note: Please do not respond with "I can't determine the exact age"; just provide the number you think is closest. |
| Basic Expr. | / | Please describe the facial emotional features of the person in the image. |
| Comp. Expr. | | |
| Deepfake | A forged face may be generated by face-swapping, which is a technique that replaces one person's facial features with those of another person. | Please analyze whether there are any artifacts indicating face-swapping in the facial image. |
| | A forged face may be generated by face-reenactment, which is a technique that transfers the facial expressions and movements of one person onto another person's face in real-time or in a recorded video. | Please analyze whether there are any artifacts indicating face-reenactment in the facial image. |
| Spoofing | A spoof face image may be printed on paper and then re-photographed. | Please analyze whether there are any clues in the facial image that indicate it was printed on paper and then re-photographed. |
| | A spoof face image may be re-photographed after being played on a video playback device. | Please analyze whether there are any clues in the facial image that indicate it was re-photographed from a video playback device. |
| Basic FR | / | Please analyze whether the two people in the images are the same person by explaining the similarities and differences in their facial features. |
| C.P. FR | Even if the two images are of the same person, there may be differences in posture. | |
| C.A. FR | Even if the two images are of the same person, there may be differences in age, meaning the two photos were taken at different ages of this person. | |
| S.L. FR | Even if the two photos are not of the same person, they may still have similar facial features. | |
| Occ. FR | To determine whether the two partially obscured photos are of the same person, it is necessary to analyze other unobscured facial areas. | |
| H. Attr. | / | Please analyze whether the characteristics described in each option of the multiple-choice question match the person in the red box, one by one. |

| L3 Ability | Hint | Task-specific CoT instruction |
|---|---|---|
| Action | / | Please analyze the actions of the person in the red box. |
| Position | / | Please analyze the relative positional relationship between the subject (marked with a red box) and the object (marked with a green box). |
| Counting | There are fewer than 10 people in the image. | Please estimate the number of people appearing in the image, including those who are occluded or incomplete. Note: Please do not say 'I cannot determine the exact number of people'; just provide the number you think is approximate. |
| | There are fewer than 100 people in the image. | |
| | There are more than 100 people in the image, but fewer than 4,000. | |
| Social Rel. | / | Please analyze the possible social relationship between the two people in the red boxes from the perspectives of relative position, posture, and facial expressions. |
| Identity | / | Please analyze the occupation of the person in the red box from the perspectives of clothing, actions, background, etc. |
| Re-ID | If two people have significant differences in posture and their faces are relatively blurry, the main basis for determining whether they are the same person is their clothing characteristics. | Please analyze whether the two people in the images are the same person by explaining the similarities and differences in their full-body features. |

## B.3 MORE DETAILS ON THE EXPERIMENTS FOR RQ2

### B.3.1 UNEXPLORED L3 ABILITIES

We explain the reasons for not conducting experiments on the remaining 5 L3 abilities as follows.

**Face/Human Attribute Recognition** These two tasks include a large number of binary classification labels (40 labels in CelebA for face and 14 labels in WIDER Attribute for human). Using evaluation protocols designed for specialist models to fully assess the performance of MLLMs would result in huge computational costs. Additionally, many attribute labels have ambiguous semantics that are difficult to define accurately, such as "attractive," "big lips," and "big nose."

**Relative Position Understanding** In the face and human understanding community, there are no specialist models specifically constructed to perceive the spatial relationships between one person to others and objects.

**Identity Reasoning** There is a lack of publicly available specialist models that perform occupation classification.

**Social Relationship Recognition** Existing evaluation datasets for specialist models, such as PIPA and PISC, are not suitable for directly evaluating the social relationship recognition abilities of MLLMs. Because many annotations in these datasets are ambiguous (it is often impossible to assert that people in an image belong to one relationship category rather than another) and semantically overlapping (for example, "couple" is a separate category, but a married couple is semantically also part of "family"). Fine-tuned specialist models can still learn to classify under unclear standards, but for zero-shot learning MLLMs, the lack of clear definitions greatly reduces performance. It is worth noting that in Face-Human-Bench, when we use PISC to construct problems for evaluating the social relationship recognition ability of MLLMs, we manually review and remove problems with ambiguous options or potentially non-unique answers, allowing us to assess accurately.

### B.3.2 EXPLORED L3 ABILITIES

We provide the prompt templates for directly evaluating L3 abilities on public datasets from the face and human community in Tables 21 to 29. During testing, the options will be randomly shuffled.

Table 21: Prompt for **Age Estimation** on UTKFace.

What is the age of the person in the image? Please answer with a number between 0 and 100.
Answer the question with a single number, and don't provide other additional explanations.

Table 22: Prompt for **Basic Expression Recognition** on RAF-DB (Basic).

What expression is on the face in the image?
A. Surprise B. Fear C. Disgust D. Happiness E. Sadness F. Anger G. Neutral
Answer with the option's letter from the given choices directly, and don't provide other additional explanations.

Table 23: Prompt for **Compound Expression Recognition** on RAF-DB (Compound).

What expression is on the face in the image?
A. Happily Surprised B. Happily Disgusted C. Sadly Fearful D. Sadly Angry
E. Sadly Surprised F. Sadly Disgusted G. Fearfully Angry H. Fearfully Surprised
I. Angrily Surprised J. Angrily Disgusted K. Disgustedly Surprised
Answer with the option's letter from the given choices directly, and don't provide other additional explanations.

Table 24: Prompt for **Deepfake Detection** on FF++.

Is there any evidence of face forgery artifacts in the picture?
A. Yes B. No
Answer with the option's letter from the given choices directly, and don't provide other additional explanations.

Table 25: Prompt for **Face Anti-Spoofing** on SiW-Mv2.

Is the face in the picture a spoof face?
A. Yes B. No
Answer with the option's letter from the given choices directly, and don't provide other additional explanations.

Table 26: Prompt for **Basic/Cross-Pose/Cross-Age/Similar-Looking/Occluded Face Recognition** on LFW/CPLFW/CALFW/SLLFW/MLFW.

Are the people in the two photos the same person?
A. Yes B. No
Answer with the option's letter from the given choices directly, and don't provide other additional explanations.

Table 27: Prompt for **Action Recognition** on HICO-DET.

Which of the following words best describes the interaction between the person in the red box and the object in the green box?
[Opions, include all actions involving the same object extracted from HICO-DET.]
Answer with the option's letter from the given choices directly, and don't provide other additional explanations.

Table 28: Prompt for **Crowd Counting** on ShTech-A.

How many people are there in the picture approximately? Please answer with a number between 0 and 4000.
Answer the question with a single number, and don't provide other additional explanations.

Table 29: Prompt for **Person Re-Identification** on Market-1501.

Are the people in the two photos the same person?
A. Yes B. No
Answer with the option's letter from the given choices directly, and don't provide other additional explanations. Only one option is correct.

# C ADDITIONAL RESULTS

## C.1 FACE-HUMAN-BENCH (ENGLISH)

We provide the visualization of the L2 and L3 results in Figures 8 to 10.

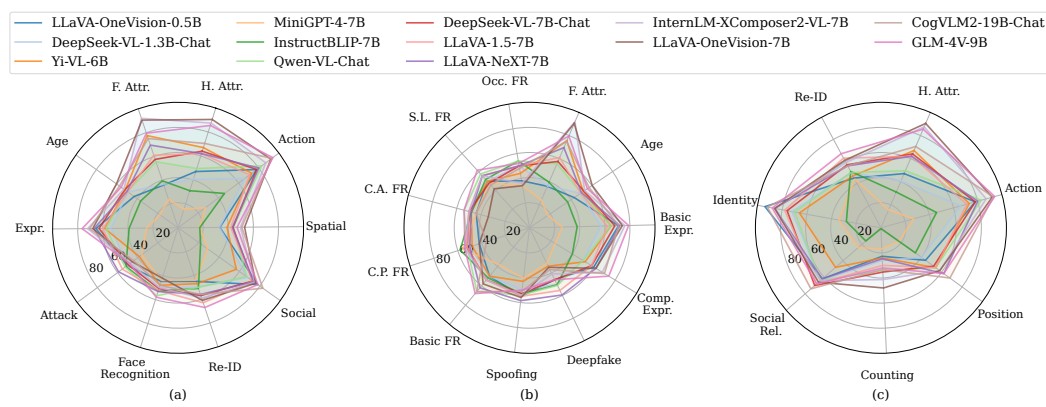

Figure 8: The performance of open-source MLLMs with LLM parameter scales below 10B on L2 and L3 abilities.

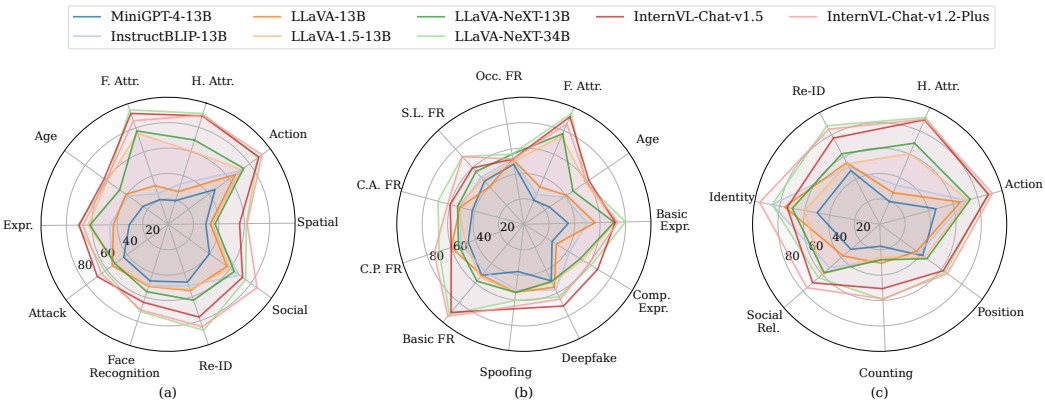

Figure 9: The performance of open-source MLLMs with LLM parameter scales above 10B on L2 and L3 abilities.

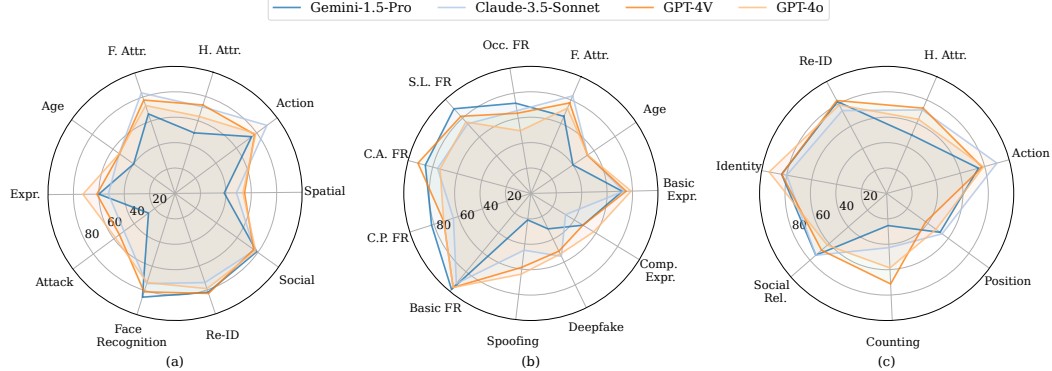

Figure 10: The performance of closed-source MLLMs on L2 and L3 abilities.

## C.2  FACE-HUMAN-BENCH (CHINESE)

Table 30 shows the performance of all evaluated MLLMs at different levels of abilities on the Human-Face-Bench (Chinese). We further compare the performance of different MLLMs on English and Chinese versions of the Face-Human-Bench, as shown in Figure 11. Models are sorted with the ascending order of average performance.

Table 30: Zero-shot scores of MLLMs on the hierarchical Face-Human-Bench (CN). The highest scores for open-source and closed-source MLLMs are marked in blue and green respectively.

| Model | | | Face Understanding | | | | | | | | | | | |
|---|---|---|---|---|---|---|---|---|---|---|---|---|---|---|
| | Attr. | Age | Expression | | | Attack Detection | | | Face Recognition | | | | | |
| | | | Basic | Comp. | Mean | DFD | FAS | mean | Basic | C.P. | C.A. | S.L. | Occ. | Mean |
| Random | 25.0 | 25.0 | 25.0 | 25.0 | 25.0 | 50.0 | 50.0 | 50.0 | 50.0 | 50.0 | 50.0 | 50.0 | 50.0 | 50.0 |
| LLaVA-OneVision-0.5B | 29.0 | 34.3 | 67.0 | 58.0 | 62.5 | 38.0 | 56.0 | 47.0 | 50.0 | 44.0 | 50.0 | 52.0 | 52.0 | 49.6 |
| DeepSeek-VL-1.3B-Chat | 37.0 | 48.7 | 61.0 | 62.0 | 61.5 | 47.0 | 50.0 | 48.5 | 50.0 | 50.0 | 48.0 | 44.0 | 50.0 | 48.4 |
| Yi-VL-6B | 60.0 | 49.3 | 67.0 | 46.0 | 56.5 | 25.0 | 28.0 | 26.5 | 36.0 | 34.0 | 34.0 | 24.0 | 38.0 | 33.2 |
| MiniGPT-4-7B | 21.0 | 21.7 | 28.8 | 25.0 | 24.0 | 50.9 | 45.5 | 39.3 | 60.4 | 57.8 | 46.7 | 35.4 | 45.7 | 45.6 |
| InstructBLIP-7B | 24.0 | 28.3 | 39.0 | 34.0 | 36.5 | 49.0 | 47.0 | 48.0 | 48.0 | 50.0 | 50.0 | 48.0 | 48.0 | 48.8 |
| Qwen-VL-Chat | 54.5 | 49.0 | 68.0 | 40.0 | 54.0 | 55.0 | 53.3 | 53.8 | 66.0 | 52.0 | 68.0 | 54.0 | 50.0 | 58.0 |
| DeepSeek-VL-7B-Chat | 67.5 | 54.7 | 65.0 | 52.0 | 58.5 | 49.0 | 51.0 | 50.0 | 58.0 | 52.0 | 40.0 | 42.0 | 42.0 | 46.8 |
| LLaVA-1.5-7B | 48.0 | 49.7 | 51.0 | 56.0 | 53.5 | 54.5 | 51.0 | 52.8 | 64.0 | 46.0 | 46.0 | 62.0 | 46.0 | 52.8 |
| LLaVA-NeXT-7B | 39.5 | 40.0 | 66.0 | 68.0 | 67.0 | 55.5 | 50.0 | 52.0 | 56.0 | 52.0 | 52.0 | 52.0 | 46.0 | 51.6 |
| InternLM-XComposer2-VL-7B | 87.0 | 53.0 | 74.0 | 68.0 | 71.0 | 45.0 | 51.0 | 48.0 | 58.0 | 46.0 | 48.0 | 66.0 | 34.0 | 50.4 |
| LLaVA-OneVision-7B | 91.0 | 61.0 | 75.0 | 60.0 | 67.5 | 35.0 | 52.0 | 43.5 | 60.0 | 38.0 | 20.0 | 36.0 | 28.0 | 36.4 |
| CogVLM2-19B-Chat | 77.5 | 55.7 | 76.0 | 68.0 | 72.0 | 40.0 | 45.0 | 42.5 | 60.0 | 40.0 | 56.0 | 68.0 | 48.0 | 54.4 |
| GLM-4V-9B | 84.5 | 58.3 | 80.0 | 78.0 | 79.0 | 37.0 | 52.0 | 44.5 | 72.0 | 60.0 | 68.0 | 70.0 | 64.0 | 66.8 |
| MiniGPT-4-13B | 18.5 | 26.0 | 35.4 | 35.4 | 33.5 | 50.8 | 43.9 | 29.0 | 52.1 | 50.0 | 60.0 | 39.5 | 51.0 | 46.8 |
| InstructBLIP-13B | 7.0 | 29.0 | 37.2 | 31.3 | 21.0 | 59.5 | 47.4 | 27.2 | 7.1 | 9.5 | 12.2 | 12.8 | 25.0 | 10.8 |
| LLaVA-13B | 24.5 | 37.7 | 56.6 | 29.4 | 34.0 | 50.8 | 54.5 | 44.0 | 52.1 | 54.0 | 52.0 | 56.0 | 46.0 | 51.6 |
| LLaVA-1.5-13B | 62.0 | 53.0 | 72.0 | 60.0 | 66.0 | 51.5 | 53.5 | 52.5 | 62.0 | 54.0 | 50.0 | 50.0 | 50.0 | 53.2 |
| LLaVA-NeXT-13B | 54.5 | 44.0 | 69.1 | 37.5 | 51.5 | 53.1 | 56.0 | 54.0 | 58.0 | 50.0 | 60.0 | 50.0 | 50.0 | 53.6 |
| InternVL-Chat-v1.5 | 89.0 | 61.3 | 82.0 | 70.0 | 76.0 | 61.0 | 62.0 | 61.5 | 94.0 | 68.0 | 62.0 | 66.0 | 48.0 | 67.6 |
| LLaVA-NeXT-34B | 93.5 | 55.3 | 83.0 | 58.0 | 70.5 | 63.0 | 63.0 | 63.0 | 92.0 | 68.0 | 78.0 | 70.0 | 58.0 | 73.2 |
| InternVL-Chat-v1.2-Plus | 87.0 | 57.3 | 73.0 | 52.0 | 62.5 | 61.5 | 60.5 | 61.0 | 96.0 | 78.0 | 68.0 | 72.0 | 48.0 | 72.4 |
| Gemini-1.5-Pro | 58.5 | 29.0 | 70.0 | 36.0 | 53.0 | 11.0 | 16.0 | 13.5 | 98.0 | 74.0 | 84.0 | 88.0 | 68.0 | 82.4 |
| Claude-3.5-Sonnet | 79.5 | 54.0 | 74.0 | 38.0 | 56.0 | 55.0 | 57.0 | 56.0 | 90.0 | 74.0 | 82.0 | 72.0 | 60.0 | 75.6 |
| GPT-4V | 68.5 | 55.0 | 75.0 | 54.0 | 64.5 | 50.0 | 54.5 | 52.3 | 90.0 | 58.0 | 84.0 | 84.0 | 68.0 | 76.8 |
| GPT-4o | 77.5 | 57.0 | 82.0 | 70.0 | 76.0 | 52.0 | 56.0 | 54.0 | 78.0 | 60.0 | 68.0 | 80.0 | 54.0 | 68.0 |

| Model | | | Human Understanding | | | | | | | Face | Human | Per. | Rea. | Overall |
|---|---|---|---|---|---|---|---|---|---|---|---|---|---|---|
| | Attr. | Action | Spatial Relation | | | Social Relation | | | Re-ID | | | | | |
| | | | RPU | CC | Mean | SRR | IR | Mean | | | | | | |
| Random | 25.0 | 25.0 | 25.0 | 25.0 | 25.0 | 25.0 | 25.0 | 25.0 | 50.0 | 35.0 | 30.0 | 29.2 | 37.5 | 32.5 |
| LLaVA-OneVision-0.5B | 37.5 | 62.0 | 42.0 | 20.0 | 31.0 | 64.0 | 82.0 | 73.0 | 51.0 | 44.5 | 50.9 | 45.4 | 51.2 | 47.7 |
| DeepSeek-VL-1.3B-Chat | 35.0 | 60.0 | 44.0 | 24.7 | 34.3 | 64.0 | 82.0 | 73.0 | 50.0 | 48.8 | 50.5 | 48.4 | 51.4 | 49.6 |
| Yi-VL-6B | 56.5 | 68.0 | 46.0 | 24.0 | 35.0 | 50.0 | 74.0 | 62.0 | 44.0 | 45.1 | 53.1 | 52.8 | 43.6 | 49.1 |
| MiniGPT-4-7B | 25.0 | 29.0 | 37.2 | 28.2 | 25.0 | 38.6 | 38.1 | 33.0 | 36.0 | 30.3 | 29.6 | 26.7 | 34.9 | 30.0 |
| InstructBLIP-7B | 30.0 | 24.0 | 28.0 | 10.0 | 17.0 | 32.7 | 45.8 | 38.0 | 51.0 | 37.1 | 32.0 | 31.8 | 38.7 | 34.6 |
| Qwen-VL-Chat | 44.0 | 72.0 | 46.0 | 26.8 | 35.7 | 46.8 | 81.6 | 62.0 | 64.0 | 53.9 | 55.5 | 54.5 | 54.9 | 54.7 |
| DeepSeek-VL-7B-Chat | 55.5 | 81.0 | 54.0 | 40.7 | 47.3 | 66.0 | 82.0 | 74.0 | 50.0 | 55.5 | 61.6 | 61.2 | 54.5 | 58.5 |
| LLaVA-1.5-7B | 35.0 | 65.0 | 30.0 | 32.9 | 31.3 | 66.0 | 88.0 | 77.0 | 64.0 | 51.3 | 54.5 | 50.7 | 56.3 | 52.9 |
| LLaVA-NeXT-7B | 33.0 | 70.0 | 28.0 | 25.2 | 26.3 | 54.0 | 92.0 | 73.0 | 55.0 | 50.0 | 51.5 | 50.3 | 51.5 | 50.7 |
| InternLM-XComposer2-VL-7B | 75.0 | 78.0 | 60.0 | 45.3 | 52.7 | 62.0 | 84.0 | 73.0 | 70.0 | 61.9 | 69.7 | 68.7 | 61.5 | 65.8 |
| LLaVA-OneVision-7B | 84.5 | 89.0 | 48.0 | 46.7 | 47.3 | 74.0 | 92.0 | 83.0 | 61.0 | 59.9 | 73.0 | 72.8 | 56.9 | 66.4 |
| CogVLM2-19B-Chat | 66.5 | 86.0 | 56.0 | 29.3 | 42.7 | 64.0 | 98.0 | 81.0 | 60.0 | 60.4 | 67.2 | 66.7 | 59.5 | 63.8 |
| GLM-4V-9B | 77.0 | 91.0 | 62.0 | 32.0 | 47.0 | 66.0 | 90.0 | 78.0 | 62.0 | 66.6 | 71.0 | 72.4 | 63.5 | 68.8 |
| MiniGPT-4-13B | 28.5 | 32.0 | 24.5 | 26.6 | 23.3 | 18.4 | 40.4 | 28.0 | 44.0 | 30.8 | 31.2 | 27.9 | 35.5 | 31.0 |
| InstructBLIP-13B | 5.0 | 41.0 | 17.0 | 7.0 | 10.0 | 42.9 | 65.2 | 48.0 | 8.0 | 19.0 | 22.4 | 21.7 | 19.2 | 20.7 |
| LLaVA-13B | 22.5 | 59.0 | 26.5 | 31.1 | 26.7 | 38.0 | 73.5 | 55.0 | 55.0 | 38.4 | 43.6 | 36.9 | 47.1 | 41.0 |
| LLaVA-1.5-13B | 38.0 | 70.0 | 24.0 | 18.0 | 21.0 | 62.0 | 88.0 | 75.0 | 61.0 | 57.3 | 53.0 | 56.9 | 52.6 | 55.2 |
| LLaVA-NeXT-13B | 47.5 | 74.0 | 40.0 | 33.0 | 35.7 | 51.0 | 84.0 | 67.0 | 58.0 | 51.5 | 56.4 | 54.3 | 53.6 | 54.0 |
| InternVL-Chat-v1.5 | 80.5 | 87.0 | 50.0 | 50.0 | 50.0 | 70.0 | 82.0 | 76.0 | 87.0 | 71.1 | 76.1 | 75.9 | 70.2 | 73.6 |
| LLaVA-NeXT-34B | 87.5 | 83.0 | 64.0 | 44.7 | 54.3 | 62.0 | 88.0 | 75.0 | 94.0 | 71.1 | 78.8 | 75.5 | 74.1 | 74.9 |
| InternVL-Chat-v1.2-Plus | 80.0 | 88.0 | 52.0 | 50.0 | 51.0 | 72.0 | 98.0 | 85.0 | 88.0 | 68.0 | 78.4 | 72.6 | 74.1 | 73.2 |
| Gemini-1.5-Pro | 46.0 | 79.0 | 52.0 | 24.7 | 38.3 | 78.0 | 78.0 | 78.0 | 49.0 | 47.3 | 58.1 | 46.5 | 61.9 | 52.7 |
| Claude-3.5-Sonnet | 55.0 | 83.0 | 50.0 | 36.7 | 43.3 | 64.0 | 78.0 | 71.0 | 78.0 | 64.2 | 66.1 | 63.9 | 67.0 | 65.1 |
| GPT-4V | 51.0 | 59.0 | 48.0 | 65.3 | 56.7 | 60.0 | 78.0 | 69.0 | 74.0 | 63.4 | 61.9 | 58.4 | 69.1 | 62.7 |
| GPT-4o | 51.0 | 74.0 | 54.0 | 51.3 | 52.7 | 70.0 | 92.0 | 81.0 | 69.0 | 66.5 | 65.5 | 64.9 | 67.7 | 66.0 |

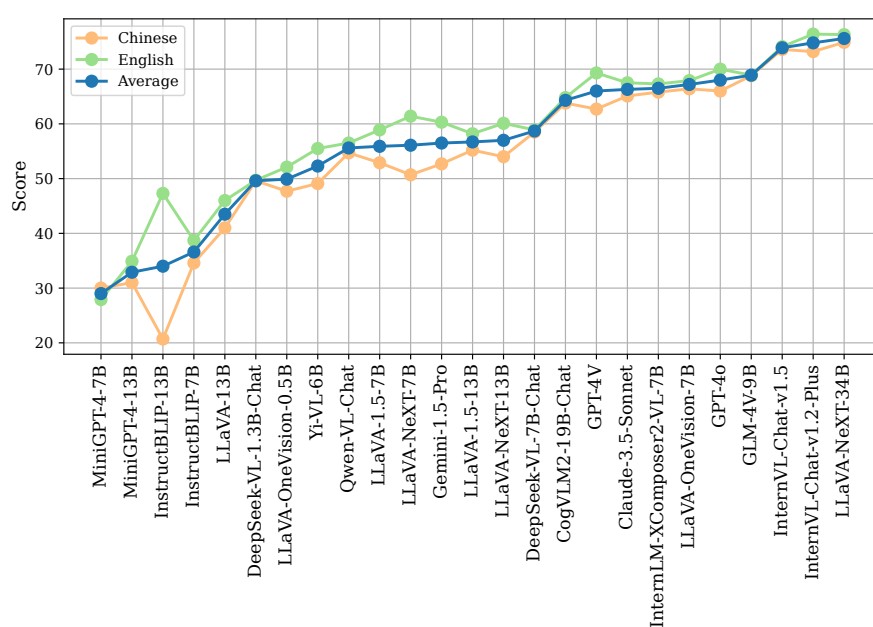

Figure 11: Comparation for the performance of different MLLMs on English and Chinese versions of the Face-Human-Bench.

## C.3 CORRELATION BETWEEN ABILITIES

The correlation coefficient matrix for L3 is shown in Figure 12. Pay particular attention to the ability correlations highlighted in the red boxes.

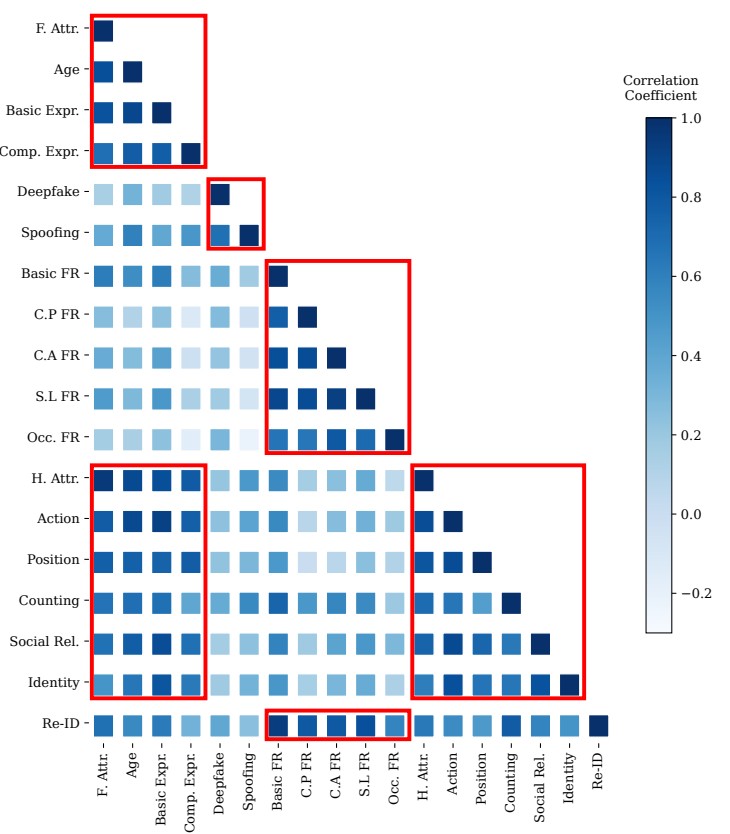

Figure 12: Correlation coefficient matrix for L3.

## C.4 RELATIVE POSITION OF TARGETS

Table 31 presents the performance differences of MLLMs across different relative positions of targets, under the three face understanding abilities and human attribute recognition.

Table 31: The impact of the relative position of targets on performance in four L3 abilities. Models with absolute performance differences greater than 5 between the two versions are highlighted in orange. Models with the smallest RPSS are marked in green.

| Model | Facial Attribute | | | Age | | | Basic Expression | | | Human Attribute | | | RPSS |
|---|---|---|---|---|---|---|---|---|---|---|---|---|---|
| | Ori. | Crop. | Dif. | Ori. | Crop. | Dif. | Ori. | Crop. | Dif. | Boxed | Crop. | Diff. | |
| LLaVA-OneVision-0.5B | 37.0 | 35.0 | 2.0 | 44.0 | 42.0 | 2.0 | 68.0 | 74.0 | -6.0 | 50.0 | 44.0 | 6.0 | 16.0 |
| DeepSeek-VL-1.3B-Chat | 35.0 | 38.0 | -3.0 | 50.7 | 47.3 | 3.3 | 58.0 | 56.0 | 2.0 | 34.0 | 47.0 | -13.0 | 21.3 |
| Yi-VL-6B | 77.0 | 74.0 | 3.0 | 55.3 | 48.0 | 7.3 | 60.0 | 70.0 | -10.0 | 59.0 | 75.0 | -16.0 | 36.3 |
| MiniGPT-4-7B | 23.0 | 25.0 | -2.0 | 16.0 | 19.3 | -3.3 | 28.0 | 24.0 | 4.0 | 18.0 | 13.0 | 5.0 | 14.3 |
| InstructBLIP-7B | 46.0 | 33.0 | 13.0 | 38.7 | 34.7 | 4.0 | 36.0 | 40.0 | -4.0 | 27.0 | 35.0 | -8.0 | 29.0 |
| Qwen-VL-Chat | 57.0 | 54.0 | 3.0 | 48.7 | 50.7 | -2.0 | 66.0 | 64.0 | 2.0 | 48.0 | 51.0 | -3.0 | 10.0 |
| DeepSeek-VL-7B-Chat | 57.0 | 58.0 | -1.0 | 52.0 | 52.7 | -0.7 | 62.0 | 74.0 | -12.0 | 55.0 | 73.0 | -18.0 | 31.7 |
| LLaVA-1.5-7B | 59.0 | 63.0 | -4.0 | 48.0 | 50.7 | -2.7 | 60.0 | 64.0 | -4.0 | 55.0 | 69.0 | -14.0 | 24.7 |
| LLaVA-NeXT-7B | 68.0 | 71.0 | -3.0 | 52.0 | 48.0 | 4.0 | 68.0 | 76.0 | -8.0 | 58.0 | 66.0 | -8.0 | 23.0 |
| InternLM-XComposer2-VL-7B | 91.0 | 93.0 | -2.0 | 52.7 | 53.3 | -0.7 | 76.0 | 76.0 | 0.0 | 87.0 | 88.0 | -1.0 | 3.7 |
| LLaVA-OneVision-7B | 91.0 | 90.0 | 1.0 | 61.3 | 59.3 | 2.0 | 72.0 | 76.0 | -4.0 | 90.0 | 91.0 | -1.0 | 8.0 |
| CogVLM2-19B-Chat | 75.0 | 75.0 | 0.0 | 59.3 | 55.3 | 4.0 | 70.0 | 72.0 | -2.0 | 67.0 | 74.0 | -7.0 | 13.0 |
| GLM-4V-9B | 83.0 | 76.0 | 7.0 | 60.0 | 51.3 | 8.7 | 80.0 | 78.0 | 2.0 | 86.0 | 85.0 | 1.0 | 18.7 |
| MiniGPT-4-13B | 19.0 | 22.0 | -3.0 | 22.7 | 26.0 | -3.3 | 34.0 | 36.0 | -2.0 | 23.0 | 16.0 | 7.0 | 15.3 |
| InstructBLIP-13B | 28.0 | 23.0 | 5.0 | 40.7 | 36.0 | 4.7 | 50.0 | 50.0 | 0.0 | 39.0 | 28.0 | 11.0 | 20.7 |
| LLaVA-13B | 35.0 | 29.0 | 6.0 | 38.0 | 43.3 | -5.3 | 52.0 | 60.0 | -8.0 | 28.0 | 26.0 | 2.0 | 21.3 |
| LLaVA-1.5-13B | 74.0 | 77.0 | -3.0 | 57.3 | 60.0 | -2.7 | 70.0 | 74.0 | -4.0 | 46.0 | 75.0 | -29.0 | 38.7 |
| LLaVA-NeXT-13B | 77.0 | 78.0 | -1.0 | 52.7 | 40.7 | 12.0 | 74.0 | 68.0 | 6.0 | 64.0 | 75.0 | -11.0 | 30.0 |
| InternVL-Chat-v1.5 | 93.0 | 91.0 | 2.0 | 63.3 | 60.0 | 3.3 | 72.0 | 72.0 | 0.0 | 87.0 | 92.0 | -5.0 | 10.3 |
| LLaVA-NeXT-34B | 96.0 | 94.0 | 2.0 | 59.3 | 58.0 | 1.3 | 82.0 | 78.0 | 4.0 | 90.0 | 93.0 | -3.0 | 10.3 |
| InternVL-Chat-v1.2-Plus | 86.0 | 86.0 | 0.0 | 61.3 | 58.0 | 3.3 | 72.0 | 76.0 | -4.0 | 88.0 | 92.0 | -4.0 | 11.3 |
| Gemini-1.5-Pro | 65.0 | 67.0 | -2.0 | 52.7 | 28.0 | 24.7 | 78.0 | 66.0 | 12.0 | 43.0 | 57.0 | -14.0 | 52.7 |
| Claude-3.5-Sonnet | 86.0 | 81.0 | 5.0 | 57.3 | 50.7 | 6.7 | 78.0 | 68.0 | 10.0 | 76.0 | 67.0 | 9.0 | 30.7 |
| GPT-4V | 79.0 | 76.0 | 3.0 | 54.7 | 52.7 | 2.0 | 76.0 | 74.0 | 2.0 | 67.0 | 79.0 | -12.0 | 19.0 |
| GPT-4o | 80.0 | 74.0 | 6.0 | 63.3 | 58.7 | 4.7 | 86.0 | 80.0 | 6.0 | 54.0 | 73.0 | -19.0 | 35.7 |

## C.5 COT PROMPTING

Based on Table 32, we explore the main reasons for the performance improvements of GPT-4o in each ability at L3, as shown in Figure 13.

Table 32: Scores of the best open-source model, InternVL-Chat-v1.2-Plus, and the best closed-source model, GPT-4o, under different settings on the hierarchical Face-Human-Bench. The highest scores for open-source and closed-source MLLMs are marked in blue and green respectively.

| Model | Setting | Face Understanding | | | | | | | | | | | | | |
|---|---|---|---|---|---|---|---|---|---|---|---|---|---|---|---|
| | | Attr. | Age | Expression | | | Attack Detection | | | Face Recognition | | | | | |
| | | | | Basic | Comp. | Mean | DFD | FAS | Mean | Basic | C.P. | C.A. | S.L. | Occ. | Mean |
| InternVL -Chat-v1.2-Plus | ZS | 86.0 | 59.7 | 74.0 | 60.0 | 67.0 | 65.5 | 65.0 | 65.3 | 94.0 | 74.0 | 62.0 | 72.0 | 52.0 | 70.8 |
| | H | 87.0 | 60.0 | 71.0 | 52.0 | 61.5 | 66.0 | 64.0 | 65.0 | 92.0 | 66.0 | 56.0 | 74.0 | 52.0 | 68.0 |
| | H+VCoT | 86.0 | 58.3 | 70.0 | 64.0 | 67.0 | 65.5 | 61.0 | 63.3 | 92.0 | 68.0 | 58.0 | 80.0 | 56.0 | 70.8 |
| | H+1TCoT | 89.0 | 61.0 | 71.0 | 50.0 | 60.5 | 58.0 | 66.0 | 62.0 | 90.0 | 68.0 | 64.0 | 76.0 | 54.0 | 70.4 |
| | H+2TCoT | 88.0 | 62.3 | 72.0 | 54.0 | 63.0 | 58.0 | 66.5 | 62.3 | 94.0 | 66.0 | 56.0 | 78.0 | 56.0 | 70.0 |
| GPT-4o | ZS | 77.0 | 61.0 | 83.0 | 62.0 | 72.5 | 53.0 | 64.0 | 58.5 | 96.0 | 72.0 | 74.0 | 76.0 | 50.0 | 73.6 |
| | H | 77.0 | 61.0 | 83.0 | 62.0 | 72.5 | 52.0 | 83.0 | 67.5 | 96.0 | 80.0 | 86.0 | 90.0 | 64.0 | 83.2 |
| | H+VCoT | 85.0 | 59.3 | 85.0 | 58.0 | 71.5 | 70.0 | 93.0 | 81.5 | 94.0 | 76.0 | 86.0 | 90.0 | 78.0 | 84.8 |
| | H+1TCoT | 89.5 | 60.7 | 84.0 | 66.0 | 75.0 | 67.0 | 94.0 | 80.5 | 98.0 | 76.0 | 84.0 | 88.0 | 72.0 | 83.6 |
| | H+2TCoT | 89.5 | 63.0 | 79.0 | 72.0 | 75.5 | 61.0 | 89.0 | 75.0 | 78.0 | 90.0 | 78.0 | 88.0 | 76.0 | 82.0 |

| Model | Setting | Human Understanding | | | | | | | | | | | | |
|---|---|---|---|---|---|---|---|---|---|---|---|---|---|---|
| | | Attr. | Action | Spatial Relation | | | Social Relation | | | Re-ID | Face | Human | Per. | Rea. | Overall |
| | | | | RPU | CC | Mean | SRR | IR | Mean | | | | | | |
| InternVL -Chat-v1.2-Plus | ZS | 90.0 | 92.0 | 66.0 | 58.7 | 62.3 | 76.0 | 86.0 | 81.0 | 85.0 | 69.7 | 83.1 | 76.7 | 76.0 | 76.4 |
| | H | 90.0 | 95.0 | 60.0 | 60.6 | 60.3 | 76.0 | 94.0 | 85.0 | 86.0 | 68.4 | 83.2 | 76.4 | 75.9 | 75.9 |
| | H+VCoT | 87.0 | 94.0 | 48.0 | 65.6 | 56.3 | 78.0 | 86.0 | 87.0 | 88.0 | 69.1 | 82.5 | 75.9 | 74.8 | 75.7 |
| | H+1TCoT | 89.0 | 92.0 | 58.0 | 51.0 | 54.3 | 74.0 | 94.0 | 84.0 | 88.0 | 68.6 | 81.4 | 75.6 | 74.3 | 75.0 |
| | H+2TCoT | 87.0 | 92.0 | 58.0 | 51.3 | 54.6 | 72.0 | 92.0 | 82.0 | 80.0 | 69.1 | 79.1 | 75.8 | 71.8 | 74.1 |
| GPT-4o | ZS | 63.5 | 81.0 | 50.0 | 58.7 | 54.3 | 66.0 | 94.0 | 80.0 | 79.0 | 68.5 | 71.6 | 68.9 | 71.7 | 70.0 |
| | H | 63.5 | 81.0 | 50.0 | 55.3 | 52.7 | 66.0 | 94.0 | 80.0 | 96.0 | 72.2 | 74.6 | 70.4 | 78.0 | 73.4 |
| | H+VCoT | 81.0 | 91.0 | 58.0 | 55.3 | 56.7 | 72.0 | 82.0 | 77.0 | 98.0 | 76.4 | 80.7 | 78.2 | 77.2 | 78.6 |
| | H+1TCoT | 81.0 | 87.0 | 60.0 | 62.7 | 61.3 | 74.0 | 90.0 | 82.0 | 98.0 | 77.9 | 81.9 | 79.0 | 81.2 | 79.9 |
| | H+2TCoT | 79.5 | 88.0 | 58.0 | 61.3 | 59.7 | 78.0 | 88.0 | 83.0 | 96.0 | 77.0 | 81.2 | 78.4 | 77.2 | 79.1 |

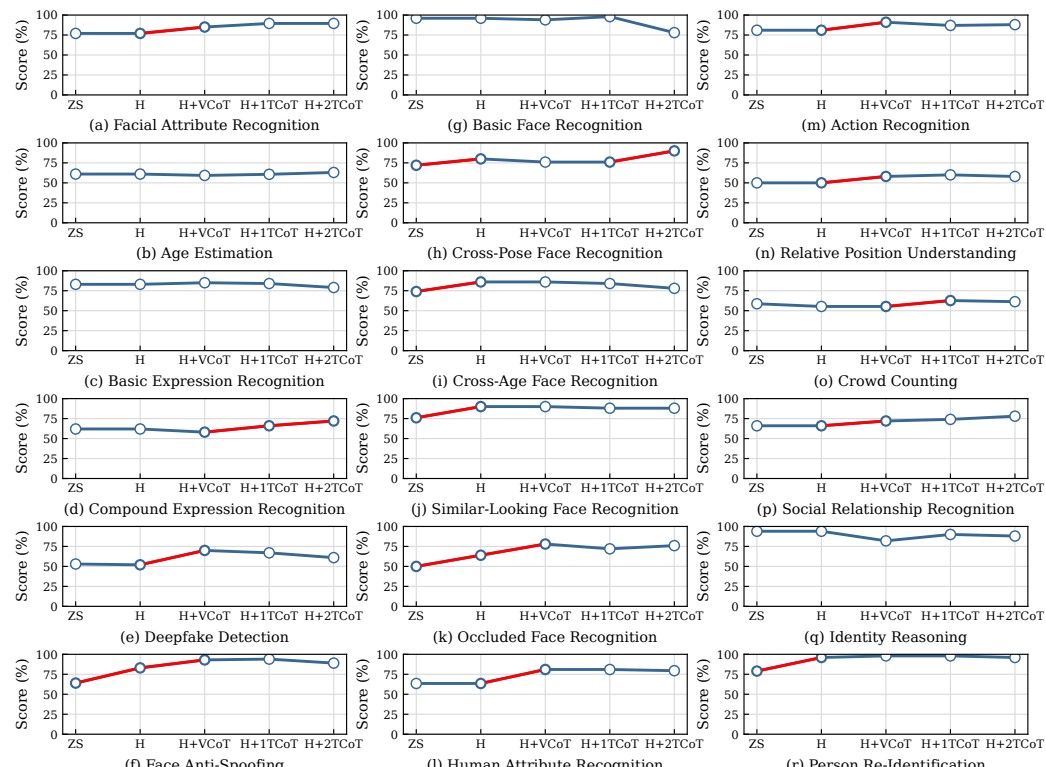

Figure 13: Main reasons of performance improvements for each L3 ability are highlighted in red.

Abilities with performance improvements mainly due to hints include face anti-spoofing, cross-pose face recognition, cross-age face recognition, similar-looking face recognition, occluded face recognition, and person re-identification.

Abilities with performance improvements mainly due to vanilla CoT instructions include facial attribute recognition, deepfake detection, face anti-spoofing, occluded face recognition, human attribute recognition, action recognition, relative position understanding, and social relationship recognition. Comparison of outputs from H and H + VCoT settings is shown in Tables 33 to 36.

Abilities with performance improvements mainly due to 1-stage task-specific CoT instructions include compound expression recognition and crowd Counting. Comparison of outputs from H + VCoT and H + 1TCoT settings is shown in Table 37.

Abilities with performance improvements mainly due to 2-stage task-specific CoT instructions include compound expression recognition and cross-pose face recognition. Comparison of outputs from H + 1TCoT and H + 2TCoT settings is shown in Table 38.

Table 33: Comparison of outputs from H and H + VCoT settings.

| Ability | Example |
|---|---|
| Facial Attribute Recognition | **Image:** 

**Question:**
Are the individuals in both images one and the same?
A. male, with black hair, wearing earrings..
B. with eyeglasses, not wearing hat, with bangs.
C. not wearing necktie, not wearing lipstick, not wearing earrings.
D. D. without eyeglasses, bald, with mouth slightly open.
**Answer:** C.
**Output from the H setting:**
B
**Output from the H + VCoT setting:**
To determine which description best matches the person in the picture, let's analyze each option based on the characteristics of the person depicted: A. male, with black hair, wearing earrings. The person in the image does not have black hair; the hair appears to be blonde or light brown. Therefore, this option does not match. B. with eyeglasses, not wearing hat, with bangs. The person in the image is not wearing eyeglasses and does not have bangs. Therefore, this option does not match. C. not wearing necktie, not wearing lipstick, not wearing earrings The person in the image is not wearing a necktie, and there is no visibility of lipstick or earrings. Hence, this description fits. D. without eyeglasses, bald, with mouth slightly open. The person in the image is without eyeglasses, but the person is not bald. He has medium-length hair. Therefore, this option does not match. Based on the analysis, the correct description that best applies to the person in the picture is: C. not wearing necktie, not wearing lipstick, not wearing earrings. |
| Deepfake Detection | **Image:** 

**Question:**
Is the presented face digitally manipulated?
A. yes.
B. no.
**Answer:** B.
**Output from the H setting:**
A
**Output from the H + VCoT setting:**
To determine if the face presented in the image is digitally manipulated, we need to analyze several aspects of the image for indicators of face-swapping or other digital alterations. The hint provided states that signs of a forged face may include inconsistencies in the blending of facial features.Here's a step-by-step analysis based on the image: 1. Facial Symmetry: Check for unnatural asymmetry which could indicate manipulation.2. Skin Tone and Texture: Look for differences in skin tone and texture that might suggest different sources for parts of the face.3. Edges and Blending: Examine the areas around the jawline, neck, and facial features for any inconsistencies or unnatural blending. 4. Lighting and Shadows: Confirm that lighting and shadows are consistent across the entire face. 5. Facial Feature Alignment: See if the eyes, nose, and mouth appear naturally aligned with regards to the facial structure.Upon examining the image: The facial features (eyes, nose, mouth) appear naturally aligned. The skin tone seems consistent across the face. There are no obvious signs of unnatural blending or asymmetry. Based on this analysis, the face does not show clear indications of being digitally manipulated. The most likely answer is: B. no. |

Table 34: Comparison of outputs from H and H + VCoT settings.

| Ability | Example |
|---|---|
| Face Anti-Spoofing | **Image:**

**Question:**
Is the facial recognition being deceived by a presentation attack?
A. yes.
B. no.
**Answer:** B.
**Output from the H setting:**
A. yes. The image appears to exhibit characteristics of a presentation attack, such as a potentially re-photographed printed face, which can deceive facial recognition systems.
**Output from the H + VCoT setting:**
To determine if the facial recognition is being deceived by a presentation attack, we should consider the hint that a spoof face image may be printed on paper and then re-photographed. Describe significant indicators that may suggest whether the image may be printed and re-photographed: 1. Quality of the image: Low-quality features such as blurriness, low resolution, or pixelation might be signs of a re-photographed image. 2. Reflection or shadows: Unnatural reflections or shadows may indicate that the image is a photograph printed on paper. 3. Distortion: Warped or distorted facial features can also be indicative of re-photographing. 4. Edges: Visible edges or borders might suggest a physical photograph in the frame.Upon observing the provided image: The image seems to be of good quality, showing clear and sharp facial features. There are no obvious signs of reflections or unnatural shadows. The facial features appear natural, without noticeable distortion. No visible edges or borders that would indicate it is a paper photograph within the frame. Based on the provided image and considerations, it does not show clear signs of being a presentation attack. Therefore, the answer to the multiple-choice question is: B. no. |
| Occluded Face Recogntion | **Image:**

**Question:**
Question: Are the individuals portrayed in each of the two pictures one and the same?
A. no.
B. yes.
**Answer:** B.
**Output from the H setting:**
A
**Output from the H + VCoT setting:**
To determine if the individuals in the two photos are the same, we need to look at some key factors that remain visible despite the face masks.
1. **Hair color and style**: Both individuals have similar hair color and styles, though this can change over time and with age.
2. **Facial features**: Look at the shape of the brow, eyes, forehead, and ears. These areas are partially visible and can offer indications of similarity.
3. **Age difference**: Keep in mind the possible age difference and how facial features can mature over the years.
Let's analyze each area:
- The individual on the left appears younger with similar hair.
- The forehead, eyes, and brow seem consistent between the two images.
- The ear shape visible in both photos appears similar.
Based on this analysis, it is most likely that the individuals portrayed in the two pictures are the same person.
Thus, the answer is:
B. yes |

Table 35: Comparison of outputs from H and H + VCoT settings.

| Ability | Example |
|---|---|
| Human Attribute Recognition | **Image:**  **Question:**
Is the facial recognition being deceived by a presentation attack?
A. He has long hair and is wearing sunglasses but is not wearing a hat.
B. He is wearing a face mask and clothes without stripes, and he is not wearing sunglasses.
C. He has long hair and is wearing a hat with a short-sleeved shirt.
D. He has short hair and is wearing a short-sleeved shirt with stripes.
**Answer:** D.
**Output from the H setting:**
C. He has long hair and is wearing a hat with a short-sleeved shirt.
**Output from the H + VCoT setting:**
To determine which description best matches the person in the red box in the picture, let's analyze the image step by step. The person in the red box: 1. Is wearing a hat or something on his head. 2. Has short hair. 3. Is wearing a short-sleeved shirt. 4. Is not wearing sunglasses. 5. Is not wearing a face mask. 6. His shirt is plain (not striped).Based on these observations: A. This option is incorrect because the person is wearing a hat and not sunglasses. B. This option is incorrect because the person has short hair, not long hair. C. This option is incorrect because the person is not wearing a face mask. D. This option is correct because the person has short hair and is wearing a short-sleeved shirt with stripes. Therefore, the best matching description is: D. He has short hair and is wearing a short-sleeved shirt with stripes. |
| Action Recognition Hint v.s. Vanilla CoT | **Image:**  **Question:**
Which choice best depicts the actions of the individual in the red box in the image?
A. holding the bicycle.
B. riding the bicycle.
C. repairing the bicycle.
D. renting the bicycle.
**Answer:** A.
**Output from the H setting:**
B. riding the bicycle.
**Output from the H + VCoT setting:**
Upon analyzing the image, the individual in the red box is seen with one foot on the ground and holding the handlebar of the bicycle, which is leaning to one side. This indicates that the person is not riding, repairing, or renting the bicycle at that moment. Therefore, the best choice that depicts the actions of the individual in the red box is: A. holding the bicycle. |

Table 36: Comparison of outputs from H and H + VCoT settings.

| Ability | Example |
|---------|---------|
| Relative Position Understanding | **Image:** 
**Question:**
Among the following options, what is the most fitting way to characterize the subject?
A. The woman is on the table.
B. The woman is behind the table.
C. The woman is under the table.
D. The woman is in front of the table.
**Answer:** B.
**Output from the H setting:**
D. The woman is in front of the table.
**Output from the H + VCoT setting:**
To determine the most fitting way to characterize the woman's location (indicated by a red box) in relation to the table (marked with a green box), let's analyze the image step by step. The options are: A. The woman is on the table. B. The woman is behind the table.C. The woman is under the table.D. The woman is in front of the table. From the image analysis: 1. Option A suggests that the woman is on the table. The image clearly shows that she is not physically on top of the table, so this option is incorrect. 2. Option B suggests that the woman is behind the table. The image shows the woman standing further away from the camera than the table, implying that the table is closer to the viewpoint than she is. This would make the woman behind the table. 3. Option C suggests that the woman is under the table. It is evident from the image that she is standing and not underneath the table, so this option is incorrect. 4. Option D suggests that the woman is in front of the table. Given that the woman is further from the camera than the table, this is incorrect because, from the perspective of the camera, she would be behind the table. Based on the analysis, the correct answer is: B. The woman is behind the table. |
| Social Relationship Recognition | **Image:** 
**Question:**
The two people in the red box in the image are most likely to have what relationship?
A. Couple.
B. Friends.
C. Professional Relation.
D. No Relation.
**Answer:** C.
**Output from the H setting:**
B. Friends.
**Output from the H + VCoT setting:**
The two people in the red box are both wearing sports uniforms, indicating that they are likely participating in the same team activity. Since they are on the same volleyball court, it suggests they have coordinated roles within the same context, typically associated with professional relations in a sports setting. Therefore, the most likely answer is: C. Professional Relation. |

Table 37: Comparison of outputs from H + VCoT and H + 1TCoT settings. Incorrect and correct analysis contents are marked in red and blue, respectively.

| Ability | Example |
|---|---|
| Compound Expression Recognition | **Image:** 

**Question:**
What emotion is the person in the photo displaying?
A. Angrily Disgusted.
B. Happily Disgusted.
C. Disgustedly Surprised.
D. Sadly Surprised.
**Answer:** D.
**Output from the H + VCoT setting:**
To determine which emotion the baby in the photo is displaying, let's analyze each of the options given:- Happily Disgusted: An emotion combining happiness and disgust, likely would show signs of smiling with a look of dislike. Sadly Surprised: An emotion combining sadness and surprise, would show signs of tears or downturned lips with wide eyes or raised eyebrows. Disgustedly Surprised: An emotion combining disgust and surprise, would show signs of wrinkled nose, possibly pulled back lips, and wide eyes. Angrily Disgusted: An emotion combining anger and disgust, would show furrowed brows, clenched teeth, and a wrinkled nose. Analyzing the baby's face: The baby's mouth is slightly open. Eyes are wide open. Eyebrows are raised.- There are no signs of a smile or frown. These features suggest a combination of surprise and possibly a hint of discomfort or unease, but there are no clear signs of happiness or anger. Given these observations, the most likely emotion the baby is displaying is: C. Disgustedly Surprised.
**Output from the H + 1TCoT setting:**
The person in the photo has their eyebrows raised and slightly together, their eyes open wide, and their mouth slightly open. These facial features generally indicate a surprised expression, which can be mixed with another emotion. Based on the available options and the facial features observed:- Eyebrows are not furrowed or lowered, which rules out anger. The eyes are wide open, suggesting surprise or sadness rather than disgust. The mouth is not forming a clear smile or frown, maintaining a neutral or surprised shape . Given these observations, the most likely emotion is D. Sadly Surprised. The other combinations do not fit well with the facial features depicted. The correct answer is: D. Sadly Surprised |
| Crowd Counting | **Image:** 

**Question:**
How many figures are shown in this pic?
A. 7.
B. 6.
C. 5.
D. 4.
**Answer:** D.
**Output from the H + VCoT setting:**
To determine the number of people in the image, let's examine it closely: 1. There is one person wearing a white shirt at the back left. 2. There is one person wearing a black shirt with text and shorts, slightly to the right of the first person. 3. There is one person in a light gray shirt, squatting and reaching for the ball. 4. There is one person in a turquoise shirt actively engaging with the person in gray. 5. There is another person, partially obscured by the trees and goalpost, sitting on the ground in the background. Counting all these, there are 5 people in the image. The correct answer is: C. 5
**Output from the H + 1TCoT setting:**
Based on the provided image, there are a total of 4 people visible. My rationale: 1. One person in the foreground with sunglasses, actively engaging with the ball. 2. One person near the goalpost, in a teal shirt and white shorts. 3. Another person standing, wearing a black shirt and shorts. 4. Another person further back, near the goalpost and the other player. Answer: D. 4. |

Table 38: Comparison of outputs from H + 1TCoT and H + 2TCoT settings. Incorrect and correct analysis contents are marked in red and blue, respectively.

| Ability | Example |
|---|---|
| Compound Expression Recognition | **Image:** 

**Question:**
What is the facial expression of the person in this image?
A. Sadly Fearful.
B. Disgustedly Surprised.
C. Angrily Disgusted.
D. Happily Surprised.
**Answer:** A.
**Output from the H + 1TCoT setting:**
The facial expression of the person in the image includes wide eyes and an open mouth showing teeth, indicative of surprise, with a general positive emotional tone. This suggests D. Happily Surprised. The facial emotional features described match the expression shown in the image, with eyebrows raised and drawn together, eyes wide open, and mouth slightly open with a downturn indicating sadness and fear. The person is showing a mixture of happiness and surprise. Thus, the correct answer is: D. Happily Surprised.
**Output from the H + 2TCoT setting:**
A. Sadly Fearful. The facial emotional features described match the expression shown in the image, with eyebrows raised and drawn together, eyes wide open, and mouth slightly open with a downturn indicating sadness and fear. |
| Cross-Pose Face Recognition | **Image:** 

**Question:**
Are the subjects in the two photographs the identical individual?
A. yes.
B. no.
**Answer:** A.
**Output from the H + 1TCoT setting:**
Based on the analysis of the two photographs, here are the similarities and differences in their facial features: Similarities: Both individuals have light-colored hair. Both have similar styles in terms of hair length and general appearance. Differences: The person in the first image has a more mature appearance and their hair is a lighter shade, almost blond. The person on the right appears younger and has a hairstyle that frames their face differently. The facial structures and expressions differ; the person on the left has a more elongated face, whereas the person on the right has a more rounded face. Taking these observations into account, it appears that they have notable differences in their appearance which suggest that they are not the same person. Therefore, the answer is: B. no.
**Output from the H + 2TCoT setting:**
Based on the analysis of the similarities and differences: Similarities 1. Hair Color and Style: Both individuals have light-colored hair with a similar longer length and some layering. 2. Facial Structure: Both have similar facial structures, particularly around the cheeks and jawlines. Differences: 1. Age Appearance: The individual in the left photograph appears older compared to the one on the right. 2. Facial Expression and Angle: The left photograph shows a front-facing angle, while the right is a side profile, which can affect the perception of facial features. 3. Makeup and Lighting: The lighting and possible use of makeup in the left photograph make it difficult to see the natural contours of the face compared to the right photograph. Taking all these factors into account, the differences in age appearance and the angles make it challenging to definitively determine if they are the same individual. However, based on the overall similarity in facial structure and hair, it seems plausible that they could be the same person at different times in their life or captured differently. Answer: A. yes. |

## C.6 SPECIALIST MODELS SIGNIFICANTLY OUTPERFORMING MLLMs

We list the early specialist models used for comparison in Table 39.

Table 39: Early specialist models used for comparison.

| Ability | Dataset | Metric | Performance | Early Specialist Model |
|---|---|---|---|---|
| Age | UTKFace | MAE | 5.47 | CORAL (Cao et al., 2020) |
| Basic Expr. | RAF-DB (Basic) | ACC | 74.20 | DLP-CNN (Li et al., 2017b) |
| Comp. Expr. | RAF-DB (Compound) | ACC | 44.55 | DLP-CNN (Li et al., 2017b) |
| Deepfake | FF++ | ACC | 82.01 | XceptionNet Chollet (2017) |
| Spoofing | SiW-Mv2 | ACER | 9.40 | SRENet Guo et al. (2022) |
| Basic FR | LFW | ACC | 99.50 | |
| C.P. FR | CPLFW | ACC | 87.47 | R50 (He et al., 2016) |
| C.A. FR | CALFW | ACC | 92.43 | + CosFace (Wang et al., 2018) |
| S.L. FR | SLLFW | ACC | 98.40 | + CASIA-WebFace (Yi et al., 2014) |
| Occ. FR | MLFW | ACC | 82.87 | |
| Action | HICO-DET | mAP | 19.81 | ConsNet (Liu et al., 2020) |
| Counting | ShTech-A | MAE | 110.20 | MCNN (Zhang et al., 2016) |
| Re-ID | Market1501 | ACC | 95.26 | LightMBN (Herzog et al., 2021) |

## D POTENTIAL BIAS FOR DEMOGRAPHIC CHARACTERISTICS

Do MLLMs contain potential biases? Specifically, do their performances vary based on the demographic characteristics of the input faces? Existing works, such as constructing the RFW (Wang et al., 2019) and BFW (Robinson et al., 2020) datasets, have explored racial biases in face recognition systems. Inspired by these works, we investigate whether MLLMs exhibit different face recognition abilities across different racial groups.

We transform face pairs from the Caucasian, African, Asian, and Indian subsets of the RFW dataset into face recognition problems similar to those in Face-Human-Bench. The test results of the three best-performing open-source models in our main experiments are presented in Table 40, revealing the racial bias of MLLMs in face recognition ability. The performance of Caucasians is the best for each model, significantly surpassing that of other racial groups. In our future work, we will systematically evaluate the performance variations of MLLMs on samples with different demographic characteristics.

Table 40: Racial bias of MLLMs. The evaluation metric used is ACC.

| Model | Caucasian | African | Asian | Indian | Mean |
|---|---|---|---|---|---|
| ResNet34+CASIA-WebFace+ArcFace | 92.15 | 84.93 | 83.98 | 88.00 | 87.27 |
| InternVL-Chat-v1.5 | 76.62 | 60.75 | 69.67 | 71.58 | 69.65 |
| LLaVA-NeXT-34B | 71.12 | 62.23 | 66.35 | 67.15 | 66.71 |
| InternVL-Chat-v1.2-Plus | 76.68 | 67.97 | 70.38 | 72.55 | 71.90 |

## E PRIVACY PROTECTION

Face-Human-Bench can also be used to evaluate privacy protection. In some scenarios, we want MLLMs to refuse to answer certain questions related to faces and humans. In such cases, lower performance on the Face-Human-Bench indicates a higher success rate in privacy protection on this information. Table 41 presents a comparison of the performance between APIs provided by OpenAI and Azure OpenAI. Note that Azure OpenAI primarily offers security and enterprise-grade services. GPT-4V and GPT-4o from Azure OpenAI show significant performance degradation in age estimation and expression recognition. Here are some example outputs:

• "I cannot determine the age of the person in the photo **with the information provided**."

Table 41: Scores of GPT-4o and GPT-4V APIs from OpenAI and Azure OpenAI.

| Model | Attr. | Age | Expression | | | Attack Detection | | | | Face Recognition | | | | |
|---|---|---|---|---|---|---|---|---|---|---|---|---|---|---|
| | | | Basic | Comp. | Mean | DFD | FAS | mean | Basic | C.P. | C.A. | S.L. | Occ. | Mean |
| GPT-4V (Azure OpenAI) | 64.5 | 34.7 | 27.0 | 0.0 | 13.5 | 48.0 | 52.0 | 50.0 | 76.0 | 54.0 | 62.0 | 66.0 | 72.0 | 66.0 |
| GPT-4V (OpenAI) | 77.5 | 53.7 | 75.0 | 48.0 | 61.5 | 50.5 | 58.5 | 54.5 | 96.0 | 72.0 | 92.0 | 82.0 | 64.0 | 81.2 |
| GPT-4o (Azure OpenAI) | 56.0 | 41.3 | 17.0 | 0.0 | 8.5 | 46.0 | 59.0 | 52.5 | 88.0 | 62.0 | 60.0 | 80.0 | 72.0 | 72.4 |
| GPT-4o (OpenAI) | 77.0 | 61.0 | 83.0 | 62.0 | 72.5 | 53.0 | 64.0 | 58.5 | 96.0 | 72.0 | 74.0 | 76.0 | 50.0 | 73.6 |

| Model | Attr. | Action | Spatial Relation | | | Social Relation | | | Re-ID | Face | Human | Per. | Rea. | Overall |
|---|---|---|---|---|---|---|---|---|---|---|---|---|---|---|
| | | | RPU | CC | Mean | SRR | IR | Mean | | | | | | |
| GPT-4V (Azure OpenAI) | 52.0 | 82.0 | 62.0 | 48.7 | 55.3 | 64.0 | 74.0 | 69.0 | 73.0 | 45.7 | 66.3 | 49.4 | 65.8 | 56.0 |
| GPT-4V (OpenAI) | 73.0 | 78.0 | 38.0 | 71.3 | 54.7 | 68.0 | 84.0 | 76.0 | 83.0 | 65.7 | 72.9 | 66.4 | 73.7 | 69.3 |
| GPT-4o (Azure OpenAI) | 64.0 | 78.0 | 46.0 | 45.3 | 45.7 | 68.0 | 84.0 | 76.0 | 79.0 | 46.1 | 68.5 | 50.1 | 68.3 | 57.3 |
| GPT-4o (OpenAI) | 63.5 | 81.0 | 50.0 | 58.7 | 54.3 | 66.0 | 94.0 | 80.0 | 79.0 | 68.5 | 71.6 | 68.9 | 71.7 | 70.0 |

- "I'm sorry, but **the image is too blurry** to make an accurate assessment of the person's age."

- "I **don't have enough visual information** from the image provided to accurately determine the emotion being expressed by the person."

- "I'm unable to determine the person's expression due to **the blurred face**. Based on the available data, I cannot select a correct answer from the provided options."

From these outputs, it can be observed that Azure OpenAI might employ security strategies such as refusing to answer or blurring images.

## F    A DEMONSTRATION OF HOW TO ENHANCE MULTI-MODAL ASSISTANT RESPONSES WITH SPECIALIST MODELS

In Figure 14, we use media forensics as an application scenario to demonstrate how specialist models can improve the response quality of a multimodal assistant. Path 1 directly uses the MLLM to generate responses, while Path 2 introduces a well-trained specialist model for deepfake detection to determine whether there are digital artifacts on the faces in the image. By using the output of the specialist model to enhance the prompt, Path 2 ultimately allows the MLLM to provide more accurate responses.

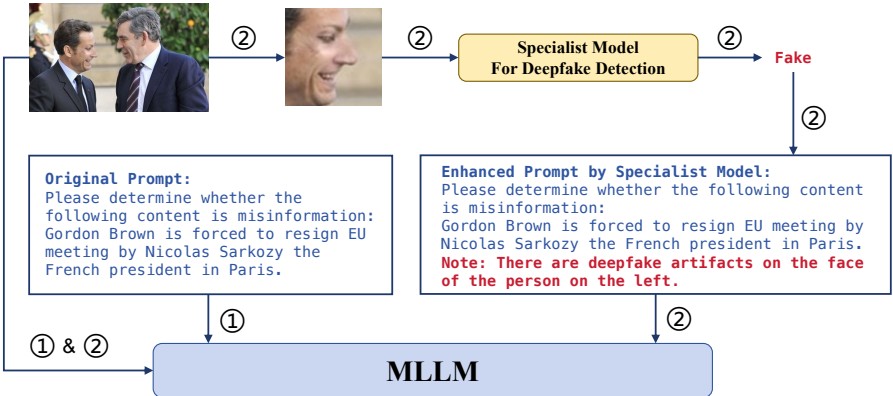

Figure 14: A demonstration of how to enhance multi-modal assistant responses with specialist models in media forensics.

## G    LIMITATIONS

Despite the rich findings, there are still some limitations in this study. (1) This is the first work to comprehensively evaluate the face and human understanding abilities of MLLMs, mainly focusing on perception and simple reasoning. It does not involve tasks that require complex reasoning by integrating multiple face and human information. We plan to explore this in future work. (2) Considering the languages supported by existing mainstream MLLMs, Face-Human-Bench currently includes only English and Chinese. The capabilities of MLLMs in understanding face and human information in more languages remain to be further explored.

# H ETHICS STATEMENT

Our work does not involve reproducing, duplicating, copying, selling, trading, reselling, or exploiting any images from the original public datasets of the face and human community for any commercial purposes. Additionally, our work does not involve further copying, publishing, or distributing any portion of the images from the original public datasets. We fully comply with the agreements of all used original public datasets.

We will only open-source the JSON files containing our test problems and the data preprocessing scripts. You need to download all the original images from the involved public datasets yourself and organize the folders according to our instructions. The data preprocessing scripts will produce images for multi-modal QAs only during testing.

In our semi-automatic data pipeline, we provide adequate compensation to all participating data reviewers and ensure that this process complies with laws and ethical guidelines. Data reviewers only remove erroneous problems and thus do not involve the impact of regional or cultural differences among reviewers.

Face-Human-Bench is intended solely for academic and research purposes. Any commercial use or other misuse that deviates from this purpose is strictly prohibited. We urge all users to respect this provision to maintain the integrity and ethical use of this valuable resource.

