# OpenReview forum: "Face-Human-Bench: A Comprehensive Benchmark of Face and Human Understanding for Multi-modal Assistants"
_ICLR.cc/2025/Conference — Submitted to ICLR 2025_

### Official Review · Reviewer_VdaD · 2024-11-02

**Soundness:** 2
**Presentation:** 3
**Contribution:** 1
**Rating:** 6
**Confidence:** 3

**Summary:**

The paper presents a study on how different Multimodal Large Language Models (MLLMs) perform on various tasks that relate to human and face understanding, and whether dedicated models can do better on those tasks than MLLMs. To this end, the authors collect in a semi-automatic way a large corpus of face and human data from existing datasets, through a manually defined curating process. The proposed Face-Human-Bench is divided following a hierarchical taxonomy of tasks from broad to fine grained tasks.

**Strengths:**

The paper presents a thorough evaluation of different models on a very well-defined, broad variety of tasks that relate human and face analysis, such as attribute classification, gender recognition, or activity recognition. The gathered corpus seems to have been curated in a neat manner and the authors are planning to make the corpus with their annotations open source. Different metrics such as accuracy and context-related performance are evaluated, shedding light on how far MLLMs are to bridge the gap w.r.t. dedicated models.


The paper is overall well presented (although it requires some proof reading and some rephrasing here and there), and the supplementary material is accompanied with multiple visual examples and the dataset collection description. I believe the results and data are of interest to the community working on exploiting LLMs for human analysis.

**Weaknesses:**

The main drawback of this paper is that it merely consists of a report. A thorough report, but that contains no technical contributions. The paper is an extensive, high-effort consistent evaluation of models, and this I believe is not enough for the paper to fit in the conference.


Section 3.3 “Correlation Between Abilities” is rather loosely written. What does the “correlation between abilities” mean? What is the measure or score that is given to an “ability” for it to be correlated with other abilities?


The RPSS as a score difference between a cropped image and the original one for attribute prediction is rather trivial and should not be counted as a contribution. That some models lose performance when prompted with local features is surprising.


I believe that the paper should also include a small description of the actual MLLMs under evaluation (or at least of the most significant ones), as well as of the data they have been trained on. Would it be possible that some of the models outperform others just because their training data was closer to that expected for human analysis? Such analysis should be included in my opinion.

**Questions:**

In my honest opinion, I believe the paper is not fit for ICLR as its contributions seem to me far from the scope of the conference. By no means I disregard the authors’ work, as it is a complete study of how Multimodal LLMs perform on a broad set of tasks regarding human analysis. However, this is all I can take from the paper, a nice, well elaborated and through study with no technical contributions. ICLR is a conference that welcomes all kinds of technical contributions within ML and CV; however, such a study I believe fits better with the IEEE Trans. on Affective Computing than on ICLR. I don’t recall seeing in ICLR over the past years such kind of report. Of course I might be wrong, and I am willing to change my mind should the authors provide evidence of former ICLR papers that are of the same kind, as this would set for precedence; or should the AC believe the paper fits within the conference.

---

> ### Author Response · Authors · 2024-11-19
>
> We express our gratitude for your valuable comments.
>
> For Weakness 1 and the Question:
>
> Over the past two years, LLMs and MLLMs have garnered significant attention from academia and industry. Unlike dedicated models designed to address a specific, well-defined task, LLMs and MLLMs exhibit strong abilities in instruction-following and reasoning, making them general-purpose assistants. An important question that arises is how to define and quantitatively evaluate the capabilities of LLMs and MLLMs. ICLR 2024 has established significant precedence for papers aiming to establish a comprehensive and scientific evaluation framework from a specific dimension for LLMs or MLLMs. Here are some papers accepted at ICLR 2024:
>
> MathVista [1] evaluates MLLMs' abilities in mathematical reasoning, SWE-bench [2] assesses LLMs' capability to resolve real-world GitHub issues, KoLA [3] explores LLMs' grasp of world knowledge, and AgentBench [4] and SmartPlay [5] evaluate LLMs as agents in challenging tasks within interactive environments. These papers, while containing no technical contributions, have enhanced our understanding of MLLM capabilities. Additionally, our paper is submitted under the primary area of “Datasets and Benchmarks”. ICLR encourages the development of high-quality datasets and benchmarks to define problems and evaluate models.
>
> [1] MathVista: Evaluating Mathematical Reasoning of Foundation Models in Visual Contexts
> https://openreview.net/forum?id=KUNzEQMWU7
>
> [2] SWE-bench: Can Language Models Resolve Real-world Github Issues?
> https://openreview.net/forum?id=VTF8yNQM66
>
> [3] KoLA: Carefully Benchmarking World Knowledge of Large Language Models
> https://openreview.net/forum?id=AqN23oqraW
>
> [4] AgentBench: Evaluating LLMs as Agents
> https://openreview.net/forum?id=zAdUB0aCTQ
>
> [5] SmartPlay: A Benchmark for LLMs as Intelligent Agents
> https://openreview.net/forum?id=S2oTVrlcp3
>
>
>
> For Weekness 2:
>
> Exploring the "correlation between abilities" aims to reveal whether the enhancement of ability A in a model is accompanied by the enhancement of ability B. Specifically, we represent the scores of all tested MLLMs on ability A and ability B from the Face-Human-Bench as X and Y, respectively. We then calculate the Pearson Correlation Coefficient to indicate the "correlation between A and B" with the following formula:
>
> $ r = \frac{{\sum (X_i - \bar{X})(Y_i - \bar{Y})}}{\sqrt{\sum (X_i - \bar{X})^2 \sum (Y_i - \bar{Y})^2}} $.
>
> For example, the correlation coefficient between age estimation and facial expression recognition is 0.88, which is close to +1.00. This indicates that enhancements in age estimation are accompanied by enhancements in facial expression recognition, and vice versa. On the other hand, the correlation coefficient between face recognition and face attack detection is 0.16, which is close to 0. This suggests that improvements in face recognition performance usually do not lead to significant gains in face attack detection.
> Theoretically, two abilities with high positive correlation might share more activation parameters within the MLLM. Conversely, two abilities with low correlation might share fewer activation parameters.
> In our updated paper submission, we will provide a clearer definition of the "correlation between abilities".
>
>
> For Weekness 3:
>
> We also do not consider that RPSS can be counted as a contribution.
> The focus of Section 3.4 is to uncover the impact of the relative position of targets on performance. When users ask MLLMs questions related to faces and humans, they do not expect the quality of the responses to fluctuate due to changes in the relative position of the targets. Our observation will inspire the MLLM community to train more robust models. RPSS is merely used as a trivial means to quantify performance fluctuations.

---

> > ### Comment · Reviewer_VdaD · 2024-11-28
> > **Comments**
> >
> > I thank the authors for the clarification in the three points raised in my review.
> >
> >
> > For W1: Thanks for the references, indeed some are actually very valuable. I understand that some of these set a precedent for assessing whether benchmarking LLMs is fit for ICLR or not. I am no one to judge if the benchmarking the reasoning capabilities of LLMs is on a similar level than benchmarking their capacity to describe facial attributes. However, it is worth noting that the latter is something already studied in the literature (e.g. https://arxiv.org/abs/2307.11760, https://ieeexplore.ieee.org/document/10195066). In that sense, it is hard to judge whether the present paper hits the bar of acceptance when compared against those mentioned above (both by myself and by the authors in their comment). In that sense, I am happy to leave such evaluation to the ACs and PCs. I am borderline with the paper in this regard, and I am happy to upgrade it to 6.
> >
> > For W2: "whether the enhancement of ability A in a model is accompanied by the enhancement of ability B" This sentence clarifies it all and I believe should be highlighted in the paper. Thanks
> >
> > For W3: I appreciate the authors' honesty on this regard; please update the manuscript to make that clear.

---

> > > ### Author Response · Authors · 2024-11-29
> > >
> > > Thank you for your valuable comments and updated scores.
> > >
> > > **For W1:**
> > >
> > > We further state the significance and contributions of our work for your reference, as follows.
> > > 1. Our work is the first to create a comprehensive benchmark for evaluating the face and human understanding abilities of MLLMs. This research can lay the groundwork for more advanced studies in complex reasoning about faces and humans. As stated in our introduction, fields such as movie understanding, multi-modal human-computer interaction, and media forensics demand higher reasoning capabilities of MLLMs about faces and humans. However, these complex reasoning capabilities are built upon a fundamental understanding of attributes, identities, and relationships. Before our work, there was a lack of systematic studies on these specific abilities of MLLMs. Our work takes the first step toward complex applications based on MLLM related to faces and humans.
> > > 2. Our evaluation extends beyond facial attributes, encompassing ten critical abilities related to faces and humans:
> > >     - Five face-related abilities: facial attribute recognition, age estimation, facial expression recognition, face attack detection, and face recognition.
> > >     - Five human-related abilities: human attribute recognition, action recognition, spatial relation understanding, social relation understanding, and person re-identification.
> > > 3. Our work differs in research subject and motivation from the two references you mentioned. Both papers you mentioned are based on the natural language domain: [1] focuses on exploring the ability of LLMs to understand emotional stimuli, and [2] focuses on emotion detection in natural language processing. In contrast, our work involves understanding both images and text, making it a multi-modal task. MLLMs must perceive visual information of faces and humans and then provide responses directly or based on simple reasoning.
> > > 4. Our study has led to numerous intriguing findings, offering feasible directions for researchers to explore the capabilities of MLLMs further. We list some of our findings as examples:
> > >
> > >       (1) For some MLLMs, when the relative position of the targets changes, the quality of the responses also varies. This insight can inspire the MLLM community to train more robust models.
> > >
> > >       (2) Under the zero-shot setting, the best closed-source model, GPT-4o, does not outperform the top-performing open-source models.
> > >
> > >       (3) Introducing hints and CoT instructions into the prompts significantly improves the performance of the open-source models but has no effect on the closed-source model. After fully leveraging its potential, GPT-4o exceeds the performance of the best open-source models.
> > >
> > > [1] Large Language Models Understand and Can Be Enhanced by Emotional Stimuli https://arxiv.org/abs/2307.11760
> > >
> > > [2] Exploring Large Language Models’ Emotion Detection Abilities: Use Cases From the Middle East https://ieeexplore.ieee.org/document/10195066
> > >
> > >
> > > **For W2:**
> > >
> > > We have added this clarification in Section 3.3 of the updated version of the paper.
> > >
> > > **For W3:**
> > >
> > > The channel for updating the PDF is currently closed. We will further emphasize this in the final version to make it clear.

---

> ### Author Response · Authors · 2024-11-19
>
> For Weekness 4:
> In Appendix B.1, we provide brief descriptions of the actual MLLMs under evaluation, but we do not include details about the data they have been trained on. We list the training data used by the three best-performing open-source models in our experiments as follows.
>
>
> |Model|Overall Score|Training Data|
> |-----|-------|----------|
> |InternVL-Chat-v1.5 |74.1 |Pre-training Stage: Laion-EN, Laion-ZH, COYO, GRIT, COCO, TextCaps, Objects, GRIT, All-Seeing, Wukong-OCR, LaionCOCO-OCR, Common Crawl PDF, MMC-Inst, LSVT, ST-VQA, RCTW-17, ReCTs, ArT, SynthDoG, COCO-Text, ChartQA, CTW, DocVQA, TextOCR, PlotQA, InfoVQA     Fine-tuning Stage: TextCaps, ShareGPTV, VQAv2, GQA, OKVQA, VSR, VisualDialog, AID, ScienceQA, TQA, ChartQA, MMC-Inst, DVQA, PlotQA, LRV-Instruction, GeoQA+, TabMWP, MathQA, CLEVR-Math/Super, Geometry3K, KVQA, A-OKVQA, ViQuAE, Wikipedia, OCRVQA, InfoVQA, TextVQA, ArT, COCO-Text, CTW, LSVT, RCTW-17, ReCTs, SynthDoG, ST-VQA, DocVQA, Common Crawl PDF, RefCOCO/+/g, Visual Genome, LLaVA-150K, LVIS-Instruct4V, ALLaVA, Laion-GPT4V, TextOCR-GPTV, SVIT,OpenHermes2.5, Alpaca-GPT, ShareGPT, COIG-CQIA
> |LLaVA-NeXT-34B|76.3|Pre-training Stage: LAION-GPT-V, ShareGPT-4V, DocVQA, SynDog-EN, ChartQA, DVQA, AI2D, RefCOCO/+/g, Toloka, LLaVA-150K, SVIT, VisDial, LRV-Instruction, LLaVA-Mix-665K     Fine-tuning Stage: COCO Caption, TextCaps, VQAv2, OKVQA, A-OKVQA, IconQA, AI2D, GQA, OCR-VQA, ChartQA, DocVQA, ST-VQA, EST-VQA, InfoVQA, LLaVAR|
> |InternVL-Chat-v1.2-Plus|76.4|LAION-en, LAION-COCO, COYO, SBU, CC3M, CC12M, Wukong, LAION-multi|
>
>
> We discuss whether some models outperform others simply because their training data was closer to that expected for human analysis, under the following 3 scenarios.
>
> (1) The categories of visual-text alignment information included in the MLLMs' training data are closer to those in Face-Human-Bench.
> This means that the training data includes more images containing faces and humans, along with detailed, accurate, and comprehensive captions for alignment. The MLLMs' abilities to understand faces and humans primarily stem from the alignment training stage (also called the pre-trained stage in some literature about MLLMs), during which visual features are aligned to the LLM's language space using visual-text pair data. During this training process, the model cannot learn language concepts it has not "seen". This "closer" is actually necessary to enhance the MLLMs' ability to understand faces and humans and is harmless to the evaluation.
>
> (2) The distribution of features in the training data about faces and humans is closer to that in the Face-Human-Bench. These features include camera angles, image resolutions, contrast, brightness, and so on.
> Since the training images used by MLLMs and the images in Face-Human-Bench come from different datasets collected separately, performance gain due to the “closer” distribution of features can be deemed negligible.
>
> (3) The training data for the MLLMs appear in the test set of Face-Human-Bench. This could lead to unfair evaluation results. We have checked the training data sources of the open-source MLLMs we tested and confirm that there is no overlap with the datasets used to construct our Face-Human-Bench. This ensures the reliability of our evaluation.

---

> > ### Author Response · Authors · 2024-11-25
> > **A Kind Reminder for Reading the Response**
> >
> > Thank you for your insightful suggestions. We have done our best to address your concerns. Since the rebuttal period is closing very soon, could you please check the response to see whether it mitigates your concerns. We would greatly appreciate that!
> >
> > Thank you for your time and consideration, the authors.

---

> > ### Comment · Reviewer_VdaD · 2024-11-28
> > **On whether some models outperform others because of training data**
> >
> > Thanks for the comment, such review I believe it is very helpful and should be properly addressed in the paper. To be clear, what is the conclusion in regards to "whether some models outperform others simply because their training data"? I understand the 3 scenarios presented by the authors, but may I ask the authors to address in which cases this is likely to be happening in the studied models?

---

> > > ### Author Response · Authors · 2024-11-29
> > >
> > > We apologize for not addressing your concern.
> > >
> > > Regarding your question: "whether some models outperform others simply because of their training data?"
> > > The straightforward answer is: Yes, such models do exist.
> > > For the LLaVA series, from LLaVA-1 to LLaVA-1.5 to LLaVA-Next, the model structure has remained unchanged. This provides us with an opportunity to observe whether the improvement in model performance can be attributed solely to changes in training data.
> > > In the table below, we list the performance of the three models at the 13B scale LLM and the corresponding training data. It can be observed that from LLaVA-1 to LLaVA-Next, the amount of training data has continuously increased. This additional data undoubtedly endows the MLLM with more world knowledge, thereby enhancing the model's understanding of faces and humans.
> > >
> > >
> > > | Model              | Overall Score on Face-Human-Bench | Training Data (The added data is marked in **bold**)                                              |
> > > |--------------------|-----------------------------------|------------------------------------------------------------------------------------------|
> > > | LLaVA-1-13B        | 46.0                                | - 558K filtered image-text pairs from LAION/CC/SBU, captioned by BLIP.                    |
> > > |                    |                                   | - 80K GPT-generated multimodal instruction-following data.                                |
> > > | LLaVA-1.5-13B      | 58.2                              | - 558K filtered image-text pairs from LAION/CC/SBU, captioned by BLIP.                    |
> > > |                    |                                   | - **158K GPT-generated multimodal instruction-following data.**                             |
> > > |                    |                                   | - **40K ShareGPT data.**                                                                  |
> > > |                    |                                   | - **450K academic-task-oriented VQA data mixture.**                                       |
> > > | LLaVA-Next-13B     | 60.1                              | - 558K filtered image-text pairs from LAION/CC/SBU, captioned by BLIP.                    |
> > > |                    |                                   | - 158K GPT-generated multimodal instruction-following data.                               |
> > > |                    |                                   | - 40K ShareGPT data.                                                                      |
> > > |                    |                                   | - **500K academic-task-oriented VQA data mixture.**                                       |
> > > |                    |                                   | - **50K GPT-4V data mixture.**                                                            |
> > > ```

---

### Official Review · Reviewer_FwuK · 2024-11-02

**Soundness:** 3
**Presentation:** 4
**Contribution:** 3
**Rating:** 6
**Confidence:** 4

**Summary:**

This paper presents a benchmark specifically designed to evaluate the facial and human understanding abilities of multimodal assistants, named Face-Human-Bench. Based on a hierarchical ability taxonomy divided into three levels, Face-Human-Bench covers 18 relevant tasks. Using the proposed Face-Human-Bench, this paper conducts a comprehensive evaluation of mainstream Multimodal Large Language Models (MLLMs) and explores the correlations between abilities, the impact of target relative positioning on MLLM performance, and the effects of Chain-of-Thought (CoT) prompting on MLLM performance.

**Strengths:**

1. The proposed Face-Human-Bench is a comprehensive evaluation benchmark that fully encompasses relevant tasks, making the assessment results more valuable and reliable.
2. The paper evaluates 25 existing mainstream MLLMs on Face-Human-Bench, with a substantial amount of experiments and rich content, intuitively demonstrating each MLLM's capabilities in facial and human understanding.
3. The paper is well-organized and clearly articulated, which improves readability and makes the findings accessible to a broad audience.

**Weaknesses:**

1. The paper mentions that Face-Human-Bench consists of a development set with 900 problems and a test set with 1800 problems, but it lacks a description of the roles of these two sets. In the subsequent experimental results, which set were the results based on?
2. There is a point of confusion regarding the calculation of the overall score: how are the weights for each sub-task determined?
3. The paper states that Face-Human-Bench supports evaluations in both English and Chinese. What insights does obtaining different results when evaluating in different languages provide? Do different models exhibit language preferences due to variations in training data?
4. The calculation method for correlation between abilities in Section 3.3 needs to be further detailed and clarified.
5. After using specialist models to assist MLLMs, did the performance of MLLMs improve on the corresponding tasks? By how much? It would be helpful to provide quantitative experimental analysis to illustrate this.
6. Minor error: In line 116, the classification of the models is reversed.

**Questions:**

See Weaknesses.

---

> ### Author Response · Authors · 2024-11-22
>
> We express our gratitude for your valuable comments.
>
> For Weakness 1:
>
> One of the purposes of constructing Face-Human-Bench is to inspire the MLLM community to train models with stronger face and human understanding capabilities. For this reason, we have divided the dataset into a development set and a test set. The experimental results reported in the paper are based on the test set. The development set is intended for the MLLM community to use in evaluating the face and human understanding abilities of MLLMs during training iterations. In our updated paper submission, we will include a description of the roles of these two sets.
>
> For Weakness 2:
>
> We set the weights of all 10 L2 abilities to be equal. For L2 abilities that encompass multiple L3 abilities, each L3 ability equally shares the weight of the corresponding L2 ability. For L3 abilities that encompass multiple image versions, each image version subset equally shares the weight of the corresponding L3 ability. Finally, we obtain the detailed weights of each subset, as shown in Table 12 at Appendix A.2.
>
> This calculation method assumes that the 10 L2 abilities have equal importance. However, in specific application scenarios, it may be more appropriate to assume different importance for each L2 ability. Table 1 provides the scores for each L3 ability, enabling readers to recalculate the overall score under different weight assumptions for the L2 abilities.
>
> For Weakness 3:
>
> As shown in the results in Appendix C.2, for nearly all the models, the test results in English outperform those in Chinese. Only a few models, such as DeepSeek-VL-1.3B-Chat, DeepSeek-VL-7B-Chat, and GLM-4V-9B, exhibit nearly identical performance in both languages.
>
> The variation in training data indeed significantly impacts the language preferences of the models. For instance, consider the models LLaVA-NeXT-7B [1] and GLM-4V-9B [2]. The former demonstrates a substantial performance gap between English and Chinese (with English outperforming Chinese by 10.7), whereas the latter shows almost identical performance in both languages (with English only slightly better by 0.1). LLaVa-NeXT-7B is based on Vicuna-7B-v1.5, which is fine-tuned from LLaMA2-7B [3]. In the pre-training corpus of LLaMA2, 89.70% is English and only 0.13% is Chinese. On the other hand, GLM-4V-9B utilizes GLM-4 [2] as the underlying LLM. Although the specific proportions of Chinese and English data are not provided, the GLM-4 technical report indicates that both Chinese and English are major languages in its pre-training data.
> This example illustrates that the proportion of different languages in the pre-training corpus of the LLM component directly affects the language preferences of the MLLM.
>
> [1] LLaVA-NeXT: Improved reasoning, OCR, and world knowledge https://llava-vl.github.io/blog/2024-01-30-llava-next/
>
> [2] ChatGLM: A Family of Large Language Models from GLM-130B to GLM-4 All Tools https://arxiv.org/abs/2406.12793
>
> [3] Llama 2: Open Foundation and Fine-Tuned Chat Models https://arxiv.org/abs/2307.09288
>
> For Weakness 4:
>
> Exploring the "correlation between abilities" aims to reveal whether the enhancement of ability A in a model is accompanied by the enhancement of ability B. Specifically, we represent the scores of all tested MLLMs on ability A and ability B from the Face-Human-Bench as X and Y, respectively. We then calculate the Pearson Correlation Coefficient to indicate the "correlation between A and B" with the following formula.
>
> $ r = \frac{{\sum (X_i - \bar{X})(Y_i - \bar{Y})}}{\sqrt{\sum (X_i - \bar{X})^2 \sum (Y_i - \bar{Y})^2}} $

---

> > ### Author Response · Authors · 2024-11-22
> >
> > For Weakness 5:
> > According to the conclusions drawn in Section 3.6, when abilities such as deepfake detection, crowd counting, and face recognition are required, it is necessary to introduce specialist models to assist MLLMs. We tested the performance of the closed-source model GPT-4o and the open-source model InternVL-Chat-v1.2-Plus on the corresponding subsets of Face-Human-Bench after incorporating specialist models.
> >
> > The test results are shown in the table below. It is evident that the performance of MLLMs significantly improves after incorporating specialist models.
> > |                                           | Deepfake Detection | Crowd Counting | Face Recognition |            |           |                 |          |       |
> > |-------------------------------------------|--------------------|----------------|------------------|------------|-----------|-----------------|----------|-------|
> > |                                           |                    |                | Basic            | Cross-Pose | Cross-Age | Similar-Looking | Occluded | Mean  |
> > | InternVL-Chat-v1.2-Plus                   | 65.5               | 58.7           | 94.0               | 74.0         | 62.0        | 72.0              | 52.0       | 70.8  |
> > | InternVL-Chat-v1.2-Plus+Specialist Models | 86.0                 | 81.3           | 100.0              | 86.0         | 88.0        | 94.0              | 82.0       | 90.0    |
> > | GPT-4o                                    | 53.0                 | 58.7           | 96.0               | 72.0         | 74.0        | 76.0              | 50.0       | 73.6  |
> > | GPT-4o+Specialist Models                  | 84.0                 | 84.7           | 98.0              | 84.0         | 88.0        | 94.0              | 76.0       | 88.0    |
> >
> > For Weakness 6:
> > Thank you for pointing out our error. The correct wording should be: "Introducing hints and CoT instructions into the prompts significantly improves the performance of the closed-source model GPT-4o, but has no effect on the open-source model, InternVL-Chat-v1.2-Plus." We will correct this mistake in our updated paper submission.

---

> > > ### Author Response · Authors · 2024-11-25
> > > **A Kind Reminder for Reading the Response**
> > >
> > > Thank you for your insightful suggestions. We have done our best to address your concerns. Since the rebuttal period is closing very soon, could you please check the response to see whether it mitigates your concerns. We would greatly appreciate that!
> > >
> > > Thank you for your time and consideration, the authors.

---

> > > ### Comment · Reviewer_FwuK · 2024-11-29
> > >
> > > Thank you for your detailed rebuttal. I appreciate your efforts in addressing my concerns, and most of them have been resolved. I’ve decided to keep my original rating. Good luck!

---

### Official Review · Reviewer_3ZXP · 2024-11-03

**Soundness:** 3
**Presentation:** 3
**Contribution:** 2
**Rating:** 3
**Confidence:** 4

**Summary:**

This paper gathers together many face and human related visual perception tasks (e.g. age and emotion recognition, crowd counting, person re-id) in the proposed Face-Human-Bench and evaluates several VLLMs (25 in total) on them.

**Strengths:**

The evaluation task pursued in this paper has some value especially for researchers working on face and human analysis. The problem is that most tasks evaluated are purely vision tasks for which many strong baselines exist. It's not surprising that specialists models outperform the VLLMs on these tasks. But arguably the authors have put a considerable amount of effort to organise the benchmark and evaluate the models. Finally, the experiment of section 3.4 is interesting.

**Weaknesses:**

Overall, unfortunately, the are a few issues with the paper which limit the impact of the proposed work:
- It's not clear whether the proposed benchmark adds something to our understanding of VLLMs.
- It's not clear why one would use a VLLM to accomplish the tasks mentioned in the paper which are visual perception tasks with very specific use cases. Since the proposed tasks are very different from the ones that the VLLMs were trained on it is perhaps not even meaningful to evaluate the models on these tasks (even the ranking of the models does not reveal some interesting information/conclusion about the VLLMs)
- In terms of novelty, the authors perform standard steps to reformulate the tasks in a manner which is understandable by the VLLM.
- Another issue is that the paper reveals very few not expected results. For example it is not surprising that sophisticated pipelines for face analysis (that perform detection, alignment and recognition) trained on datasets developed for these tasks would perform a lot better than the evaluated VLLMs  on the corresponding tasks. Nor it is surprising that models with more parameters perform better.

**Questions:**

See weaknesses above.

---

> ### Author Response · Authors · 2024-11-21
>
> We express our gratitude for your valuable comments.
>
> For Weakness 2:
>
> The purpose of our study is not to use VLLM to complete visual perception tasks. Rather, the aim is to advance the development of VLLM. An excellent VLLM must possess strong face and human understanding capabilities since images containing faces and humans are among the most common inputs for VLLM. As described in the introduction of our paper, "Faces and humans are central to social interaction, a deep understanding of this information can make multi-modal assistants achieve improved response quality and broadened application scope." A comprehensive and scientific evaluation of VLLM's face and human understanding abilities is the foundation for driving VLLM's performance improvements in this area.
>
> Under the zero-shot setting, the best closed-source model, GPT-4o, does not outperform the top-performing open-source models. This is, in fact, a rather surprising result. Our further experiments in Section 3.5 reveal that introducing hints and CoT instructions into the prompts significantly improves the performance of the open-source models but has no effect on the closed-source model. After fully leveraging its potential, GPT-4o does indeed exceed the performance of the best open-source models.
>
> For Weakness 3:
>
> Despite following standard procedures, we have constructed the first comprehensive and scientific benchmark for evaluating MLLM's face and human understanding abilities, featuring a three-level ability taxonomy.
>
> For Weaknesses 1 and 4:
>
> In fact, through a comprehensive evaluation of VLLMs' face and human understanding abilities, we have uncovered several interesting conclusions that enhance our understanding of VLLMs. For instance, for some VLLMs, when the relative position of the targets changes, the quality of the responses also varies. This insight can inspire the MLLM community to train more robust models.
>
> Another surprising result is that GPT-4o, in a zero-shot setting, does not surpass the best-performing open-source VLLMs. This has already been discussed in "For Weakness 2."
>
> Experiments in Section 3.6 suggest that sophisticated pipelines for face analysis (that perform detection, alignment and recognition) , trained on dedicated datasets, generally outperform VLLMs evaluated on corresponding tasks. However, the primary goal of this section is to provide valuable insights for constructing multi-modal agents and enhancing the overall performance of multi-modal assistants. We have comprehensively discussed the abilities in which current MLLMs significantly lag behind specialist models, as well as the abilities in which they are comparable or nearly on par.
>
> It's also worth noting that models with more parameters don't always perform better. As mentioned in Section 3.2, LLaVA-OneVision0.5B, with only 0.5 billion parameters in the LLM, outperforms the earlier InstructBLIP-13B. Additionally, the 13B versions of LLaVA-1.5 and LLaVA-Next perform slightly worse than their 7B counterparts.

---

> > ### Author Response · Authors · 2024-11-25
> > **A Kind Reminder for Reading the Response**
> >
> > Thank you for your insightful suggestions. We have done our best to address your concerns. Since the rebuttal period is closing very soon, could you please check the response to see whether it mitigates your concerns. We would greatly appreciate that!
> >
> > Thank you for your time and consideration, the authors.

---

> ### Author Response · Authors · 2024-12-01
> **A Kind Reminder for Reading the Response**
>
> Since the rebuttal period is ending soon, we would be deeply appreciative if you could review our response to ensure it addresses your points. Your feedback is invaluable to us.
>
> Thank you so much for your time and consideration.
>
> The authors

---

### Official Review · Reviewer_8qRW · 2024-11-04

**Soundness:** 4
**Presentation:** 3
**Contribution:** 4
**Rating:** 8
**Confidence:** 5

**Summary:**

The paper proposes Face-Human-Bench, a hierarchical benchmarking framework aimed at evaluating the ability of multi-modal large language models (MLLMs) to understand and interpret faces and human figures. The framework categorizes tasks across multiple levels of comprehension, from facial attributes and actions to complex reasoning like social relationships and spatial relationships. It comprises 900 development set problems and 1,800 test problems, supporting both English and Chinese. By leveraging a semi-automatic data pipeline, the authors collect and annotate images from multiple datasets to form this comprehensive benchmark. Evaluations on 25 MLLMs reveal insights on model performance relative to the task type, the relative positioning of target objects, and the effectiveness of Chain-of-Thought (CoT) prompting. Results suggest areas where specialist models surpass MLLMs, emphasizing the benchmark’s role as a holistic tool to gauge MLLM performance on face and human understanding.

**Strengths:**

- **Comprehensive Benchmarking Framework**: The face-human bench spans many abilities, providing a holistic evaluation of multimodal assistants’ capabilities in the face and human understanding.
- **New Metrics and Evaluation Protocols**: The paper introduces RPSS to measure sensitivity to the relative position of targets and percentile recall to assess retrieval in large galleries. These metrics provide nuanced insights, aiding model development.
- **Multi-Language Support**: The benchmark ensures broader applicability across language barriers by supporting both English and Chinese.
- **Empirical Findings on MLLM Performance**: The evaluation of 25 MLLMs is thorough, providing insights into model performance across diverse tasks and the potential utility of Chain-of-Thought prompting.

**Weaknesses:**

- **Limited Discussion on Dataset Biases**: Although the benchmark includes diverse tasks, the paper could expand on potential biases in the benchmark datasets, especially considering the variability in demographic representations in face and human recognition tasks.
- **Generalizability to Other Tasks**: The applicability of Face-Human-Bench to tasks beyond face and human recognition remains unclear. Expanding on how these benchmarks might generalize to other domains would add depth.
- **Impact of CoT Prompting is Limited for Open-Source Models**: While Chain-of-Thought prompting improves closed-source models, the performance gain is minimal for open-source counterparts, indicating limitations in the broader applicability of CoT.

**Questions:**

1. Dataset Composition and Diversity: Could you provide more details on the demographic diversity of the benchmark datasets, particularly for face recognition tasks?
2. Transferability Across Domains: Do you envision Face-Human-Bench or its methodologies applying to multi-modal tasks outside of face and human recognition?
3. Insights on CoT Prompting Performance: Do you hypothesize that the limited effectiveness of CoT prompting on open-source models is due to training data limitations, model architecture, or another factor?

---

> ### Author Response · Authors · 2024-11-20
>
> We express our gratitude for your valuable comments.
>
> For Question 1:
> We first utilize an off-the-shelf face detection model to extract faces from the test set of Face-Human-Bench (excluding samples for the crowd counting ability). Then, gender, age, and race labels are predicted for each face with a series of off-the-shelf specialized models. Based on these labels, we can obtain the following statistical results to demonstrate the demographic diversity of the Face-Human-Bench dataset.
> |                                                                          | Gender |        | Age  |       |       |     | Race      |         |       |        | Total  |
> |--------------------------------------------------------------------------|--------|--------|------|-------|-------|-----|-----------|---------|-------|--------|--------|
> |                                                                          | Male   | Female | 1-19 | 20-39 | 40-69 | 70+ | Caucasian | African | Asian | Indian |        |
> | Face Undersatanding                                                      | 669    | 381    | 57   | 619   | 341   | 33  | 737       | 99      | 198   | 16     | 1050   |
> | Human Undersatanding (Excluding samples for the crowd counting ability.) | 854    | 343    | 152  | 965   | 80    | 0   | 768       | 93      | 187   | 149    | 1197   |
>
> To further investigate the impact of dataset biases on the performance of MLLMs, we transform face pairs from the Caucasian, African, Asian, and Indian subsets of the RFW dataset to problems for face recognition similar to those in Face-Human-Bench.
> The test results of the three best-performing open-source models in our main experiments are presented in the following table, revealing the racial bias of MLLMs in face recognition ability. The performance for Caucasians is the best for each model, significantly surpassing that for other racial groups.
>
> |                                | Caucasian | African | Asian | Indian | Mean   |
> |--------------------------------|-----------|---------|-------|--------|--------|
> | ResNet34+CASIA-WebFace+ArcFace | 92.15     | 84.93   | 83.98 | 88.00  | 87.27  |
> | InternVL-Chat-v1.5             | 76.62     | 60.75   | 69.67 | 71.58  | 69.65  |
> | LLaVA-NeXT-34B                 | 71.12     | 62.23   | 66.35 | 67.15  | 66.71  |
> | InternVL-Chat-v1.2-Plus        | 76.68     | 67.97   | 70.38 | 72.55  | 71.90  |
>
> For Question 2:
>
> The methodologies employed to construct the Face-Human-Bench can be summarized as follows:
> 1. Review existing literature and establish a hierarchical ability taxonomy.
> 2. For each ability in the hierarchical taxonomy, collect and process relevant data. Construct a new benchmark with standardized test problems.
> 3. Identify the MLLMs to be tested and conduct testing and analysis on the new benchmark.
>
> These methodologies can be extended to evaluate other capabilities of MLLMs, such as flora and fauna understanding, or multimodal mathematical reasoning.
>
> For Question 3:
>
> In Section 3.5, we analyze the reasons behind the limited effectiveness of CoT prompting on open-source models. Despite incorporating CoT instructions into the prompts, these models fail to provide rationales in their responses. This shortcoming indicates a lack of generalization capabilities, hindering their understanding of CoT instructions.
> One possible explanation for this deficiency is the models' insufficient exposure to complex reasoning data during training, which leads them to generate direct answers without valid rationales. Another possibility is that these models have inadequate model capacity. The relatively lower number of parameters in their LLM modules constrains their reasoning abilities.

---

> > ### Author Response · Authors · 2024-11-25
> > **A Kind Reminder for Reading the Response**
> >
> > Thank you for your insightful suggestions. We have done our best to address your concerns. Since the rebuttal period is closing very soon, could you please check the response to see whether it mitigates your concerns. We would greatly appreciate that!
> >
> > Thank you for your time and consideration, the authors.

---

> > > ### Comment · Reviewer_8qRW · 2024-11-27
> > >
> > > Updated score! Excellent work, and best of luck!

---

> > > > ### Author Response · Authors · 2024-11-29
> > > >
> > > > Thank you for your recognition!
> > > >
> > > > We have updated the manuscript to add a section in the appendix to discuss the potential bias regarding demographic characteristics and referenced the BFW dataset to provide a more complete story.

---

> > ### Comment · Reviewer_8qRW · 2024-11-27
> > **Response to Rebuttal**
> >
> > Thank you for your detailed responses and clarifications regarding the demographic diversity, methodology generalizability, and CoT prompting analysis in Face-Human-Bench. I appreciate the effort taken to address the questions thoroughly.
> >
> > * * *
> >
> > #### On Question 1: Dataset Demographic Diversity
> >
> > Your response offers a detailed breakdown of the dataset's demographic distribution and highlights the bias in MLLMs using the RFW-derived experiments. While the results are insightful, they raise the following points for consideration:
> >
> > 1. **Demographic Representation**:
> > * The dataset appears to have reasonable diversity across gender, age, and racial groups. However, the relatively low representation of certain groups, such as Indian individuals in the Face Understanding tasks (16 out of 1050) and African individuals in Human Understanding (93 out of 1197), might still contribute to underperformance for these groups. Further analysis of how this impacts specific task results within the benchmark would strengthen your conclusions.
> > 2. **Bias Analysis**:
> > * The RFW-derived experiments confirm racial bias in MLLMs, with Caucasians performing significantly better. However, the racial performance disparity is stark (e.g., a nearly 10% gap between Caucasians and other groups). This could merit further discussion on potential mitigations, such as rebalancing the dataset or weighting performance metrics to emphasize fairness.
> > 3. **Comparative Evaluation**:
> >
> > * It would be helpful to explicitly link these diversity statistics to performance breakdowns within the benchmark, beyond RFW, to provide a clearer picture of how the diversity in Face-Human-Bench affects model evaluations across tasks.
> > * Consideration, whether in an actual experiment or as part of the analysis, should consider the BFW dataset, which offers a simple experiment of facial verification using balanced data. It would yield a more complete story to add to the bibliography:
> >
> > Robinson, Joseph P., et al. "Face recognition: too bias, or not too bias?." *Proceedings of the ieee/cvf conference on computer vision and pattern recognition workshops*. 2020.
> >
> >
> >
> > * * *
> >
> > #### On Question 2: Methodology Generalizability
> >
> > Your response outlines a solid methodology for extending the hierarchical ability taxonomy to other domains. This clarity is valuable, and the proposed extensions (e.g., flora/fauna understanding or mathematical reasoning) seem logical. However, for practical applicability:
> >
> > 1. **Guidelines for Extension**:
> >
> > * Detailed, reusable guidelines for data collection, problem formulation, and standardization would make this methodology more accessible to researchers aiming to replicate or extend your benchmark framework.
> > 2. **Bias Consideration in New Domains**:
> >
> > * If the work extends to other capabilities, addressing potential biases (e.g., geographic distribution of flora/fauna or cultural biases in mathematical reasoning datasets) should be a central concern. This aligns with the identified issues in demographic diversity and fairness for the Face-Human-Bench.
> >
> > * * *
> >
> > #### On Question 3: CoT Prompting for Open-Source Models
> >
> > Your analysis of CoT prompting's limited effectiveness on open-source models highlights critical generalization and model capacity issues. The reasoning provided is clear and well-supported by Section 3.5, but further elaboration on potential improvements would enhance the discussion:
> >
> > 1. **Exposure to Complex Reasoning Data**:
> >
> > * It may be helpful to explore how curated pretraining datasets emphasizing CoT reasoning could enhance these models' abilities. Additionally, providing insights into any experiments (e.g., fine-tuning on CoT-style tasks) would demonstrate the feasibility of mitigating this limitation.
> > 2. **Model Capacity Constraints**:
> >
> > * While lower parameter counts are identified as a limitation, it would be useful to contextualize this issue with examples. For instance, how do parameter counts correlate with CoT performance across open-source and proprietary models? This could help readers gauge whether scaling up model sizes is a plausible solution or whether architectural innovations might be required.
> > 3. **Alternative Techniques**:
> >
> > * Consider exploring alternatives to CoT prompting, such as retrieval-augmented generation or task-specific fine-tuning, which might bypass some current constraints on open-source models.
> >
> > * * *
> >
> > ### Summary
> >
> > Your responses address the key concerns thoroughly, providing strong evidence of the benchmark's robustness and potential for extension. The additional data on demographic diversity and bias, combined with your analysis of CoT prompting limitations, are valuable contributions. However, further clarity on mitigation strategies for demographic biases and CoT prompting challenges would strengthen the paper's impact and practical relevance.
> >
> > Thank you again for your efforts in addressing these concerns. The additional insights greatly enhance the value of your work.

---

> > > ### Author Response · Authors · 2024-11-29
> > >
> > > Thank you for your constructive comments.
> > > We will further respond to your concerns.
> > >
> > > ---
> > >
> > > #### **On Question 1: Dataset Demographic Diversity**
> > >
> > > 1. **Demographic Representation.**
> > > For human understanding tasks, the performances primarily depend on the model's comprehension of full-body attributes or the spatial and social relationships between individuals. Thus, the relatively low representation of certain groups doesn't significantly impact the results of these specific tasks. However, for face understanding tasks, the racial imbalance can indeed affect the evaluation of models for facial attributes, age, expressions, authenticity, and identity.
> > > In the supplemental experiments with RFW, we have already demonstrated that changes in race cause variations in the model's ability to understand identity. Currently, there is a lack of suitable datasets to conduct similar evaluations for facial attributes, age, expressions, and authenticity. We will develop the corresponding evaluations in our future work.
> > > 2. **Bias Analysis.**
> > > In the RFW-derived experiments, we can calculate the sum of the absolute differences obtained for each pair of racial groups. This metric can be used as a fairness score for MLLMs in face recognition. A lower score indicates better fairness. This score will inspire the MLLM community to pay attention to racial bias in the face recognition ability of models, thereby encouraging efforts to mitigate this issue when training new models.
> > > 3. **Comparative Evaluation.**
> > > (1) Thank you for your suggestion. In our future work, we will develop more scientifically sound datasets to systematically evaluate the performance variations of MLLMs on samples with different demographic characteristics. We will also conduct more analyses similar to the RFW-derived experiments.
> > > (2) We have added a section in the appendix (as Section D in the updated document) to discuss the potential bias for demographic characteristics. In this section, we reference the BFW dataset to yield a more complete story.
> > >
> > > #### **On Question 2: Methodology Generalizability**
> > >
> > > 1. **Guidelines for Extension.**
> > > Thank you for your suggestion. After the paper is published, we will open-source the code used for data collection, problem formulation, and standardization during the construction of the Face-Human-Bench, and provide relevant guidelines. This will facilitate researchers in other domains to replicate our benchmark framework.
> > > 2. **Bias Consideration in New Domains.**
> > > We have already included the bias analysis in the appendix of our updated version (Section D), which will serve as a reference for researchers in new domains.
> > >
> > > #### **On Question 3: CoT Prompting for Open-Source Models**
> > >
> > > Your suggestions are highly constructive. Investigating whether fine-tuning on CoT-style tasks can effectively improve the response quality of open-source models when faced with CoT instructions; examining whether increasing the parameters can enhance MLLM's CoT performance; and exploring whether other techniques (such as retrieval-augmented generation) can compensate for the deficiencies in CoT capabilities are all important issues for the MLLM community. Each of these issues warrants dedicated work and in-depth research, which goes far beyond the scope of our current study. Unfortunately, we currently lack the necessary resources (computational resources, funding) to conduct these investigations. We will consider these issues in our future work.
> > >
> > > ---
> > >
> > > #### **Summary**
> > >
> > > (1) We have added a section in the appendix to discuss the potential bias for demographic characteristics and referenced the BFW dataset to yield a more complete story.
> > >
> > > (2) In our future work, we plan to develop more scientifically sound datasets to systematically evaluate the performance variations of MLLMs on samples with different demographic characteristics.
> > >
> > > (3) Several important issues related to the CoT capabilities of MLLMs go far beyond the scope of our current work. Due to resource constraints (computational resources, funding), we are currently unable to provide additional conclusions about the limitations of CoT prompting. We will consider these issues in our future work.

---

### Meta-Review · Area_Chair_AmfD · 2024-12-20

**Metareview:**

The paper offers a comprehensive assessment of various models across a wide array of tasks related to human and face analysis, including attribute classification, gender recognition, and activity recognition. The corpus used appears to be meticulously curated, and the authors intend to make it, along with its annotations, openly available. Various metrics, such as accuracy and context-related performance, are analyzed to evaluate how close multimodal large language models (MLLMs) are to matching the capabilities of specialized models.
Reviewers have different opinions towards this paper. 	AC appreciates this paper's efforts to extensive experiments. However, the overall paper acts as a report without introducing significant technical novelties or theoretical contributions. Several key findings reported, such as the out-performance of open-source models over proprietary models like GPT-4 on specific tasks, lack broader relevance or significant implications for the research community. The introduction of RPSS as a novel metric is critiqued for its simplicity and trivial nature, questioning the paper’s contribution in terms of advancing evaluation methodologies. Additionally, the benchmarks’ reformulation for VLLMs is straightforward, which further highlights the paper’s lack of innovative technical content.

**Additional Comments On Reviewer Discussion:**

The authors diligently addressed reviewers’ concerns about technical contribution, novelty and relevance, as well as dataset analysis and bias issues. However, the core criticisms from Reviewer 3ZXP and Reviewer VdaD regarding the lack of technical novelty and the paper’s report-like nature were significant. Despite the authors’ attempts to justify their work, the paper failed to reach the level of innovation typically required for ICLR.

---

### Decision · Program_Chairs · 2025-01-22

Reject